# CONTRASTIVE GRAPH AUTOENCODER FOR GEOMETRIC POLYGON RETRIEVAL FROM BUILDING DATASETS

## ABSTRACT

Retrieval of polygon geometries with similar shapes from maps is a challenging geographic information task. Existing approaches can not process geometry polygons with complex shapes, (multiple) holes and are sensitive to geometric transformations (e.g., rotation and reflection). We propose Contrastive Graph Autoencoder (CGAE), a robust and effective graph representation autoencoder for extracting polygon geometries of similar shapes from real-world building maps based on template queries. By leveraging graph message-passing layers, graph feature augmentation and contrastive learning, the proposed CGAE embeds highly discriminative latent embeddings by reconstructing graph features w.r.t. the graph representations of input polygons, outperforming existing graph-based autoencoders (GAEs) in geometry retrieval of similar polygons. Experimentally, we demonstrate this capability based on template query shapes on real-world datasets and show its high robustness to geometric transformations in contrast to existing GAEs, indicating the strong generalizability and versatility of CGAE, including on complex real-world building footprints.

## 1 INTRODUCTION

Geometric shape matching and retrieval is a non-trivial task in geographic information systems especially challenging when handling complex geometries. Queries describing the geometric shapes of objects to be matched (templates) are encoded and models then retrieve objects with shapes that are visually similar. Traditional approaches encode query shapes into vector representations to search for similar geometries in databases. Statistically, Goodall (1991) presented a model-based Procrustes approach to analysing sets of shapes. Conceptually, Egenhofer (1997) first proposed a system that converts sketched queries into topological scene descriptions. Walter & Fritsch (1999) proposed a statistical approach to compute the geometric distributions (i.e., length, angle and distance) of road networks for data matching. Other methods compute geometric representations of polygons with shape contexts (Belongie et al., 2002), turning functions (Arkin et al., 1991), shape compactness (Li et al., 2013) or Fourier-transform methods (Ai et al., 2013). For instance, Xu et al. (2017) proposed a method that uses position graphs to describe geometric properties of polygons, incl. polygons with holes. Their method uses Fourier descriptors to measure shape similarity. Such methods statistically characterize the geometric information of polygons for shape matching and retrieval, but are often limited in applications to simple polygon geometries and do not generalize well to large-scale polygon databases.

Recently, learning-based methods have been developed to encode polygon geometric features for geospatial applications. van 't Veer et al. (2018) proposed deep neural networks for the semantic classifications of polygons that vectorize vertices of simple polygon boundaries as vertex sequences. A 1D convolutional neural network is applied to learn discriminative geometric features (i.e., building types). The classification accuracy of deep convolutional approaches outperformed traditional "shallow" machine learning methods (i.e., logistic regression, support vector machine (SVM) and decision tree), as shown in their study.

More recently, Mai et al. (2023) have developed a polygon encoding model that leverages non-uniform Fourier transform (NUFT) (Jiang et al., 2019) and multilayer perceptron (MLP) to generate latent embeddings of simplex-based signals from spectral feature domains. The latent embeddings of polygons in spectral domains, capturing the global structure, are applied for classifying geomet-

ric shapes and predicting spatial relations of polygons. Yan et al. (2021) proposed a graph-based convolution autoencoder to learn the compact embeddings of simple polygons for shape coding and retrieval tasks on a building footprint dataset. They represented simple polygons as graphs, with boundary vertices as nodes and the adjacency of the vertices along the outer polygon boundary as edges. They then devised a graph convolution autoencoder (GAE) to learn the normalized Laplacian features of graphs in spectral domain (Kipf & Welling, 2017), which implicitly reflect the convolutional features of $k$-hop neighboring nodes in graphs. While the GAE is effective on certain polygon retrieval tasks, it is not robust to geometric transformations (e.g., reflection and rotation) of polygon shapes and has not been tested on geometries with holes. Hence, a robust method for retrieving complex polygons with holes from spatial databases (e.g., polygonal building maps) operating on graph representations of polygons is worth investigating. We hypothesize that the robustness and strong generalisability to complex polygons (with holes) retrieval from large-scale spatial geometry databases can be achieved by a graph autoencoder leveraging the structural information and connectivity of graph representation of polygons with contrastive learning.

We thus propose an unsupervised Contrastive GAE for polygon shape matching and retrieval on large-scale geometry (e.g., building polygon) databases. In contrast to traditional models and state-of-art learning-based graph autoencoders, our Contrastive Graph Autoencoder (CGAE) is **independent of polygonal vertex counts**; capable of retrieving **polygons with or without holes**; robust to polygon **reflections and rotations**; and can effectively **generalizes** to large polygon datasets.

## 2 Graph Autoencoder

Autoencoders (Hinton & Zemel, 1993) are unsupervised learning models designed to reconstruct input data from compact embeddings in latent space. A range of autoencoders has been proposed for graph data: in particular, the variational graph autoencoder (VGAE) (Kipf & Welling, 2016) applies graph convolution layers (Kipf & Welling, 2017) as encoders to learn graph topology embeddings for link predictions. Following VGAE, the adversarially regularized graph autoencoder (Pan et al., 2018) introduced an adversarial training scheme to enforce latent embeddings of graph encoders to match a prior normal distribution and reinforce the robustness of latent embeddings. For unsupervised learning of universal latent embeddings across data domains, contrastive learning (Chen et al., 2020) has been combined with graph autoencoders (You et al., 2020; Hou et al., 2022). Graph autoencoders are thus far mainly applied on relatively small graph datasets for link prediction and unsupervised graph clustering. The applications of graph autoencoders to spatial geometries (e.g., large-scale building footprints) with contrastive learning has not been fully investigated.

Conceptually, the objective of a graph autoencoder is to learn compact latent embeddings $Z$ for input graph $G = \{A, X\}$, where $X \in \mathbb{R}^d$ denotes a node feature matrix of graph $G$, and $A$ is a square adjacency matrix that encodes the topological information of graph $G$, where $a_{i,j} \in A = 1$ if there exists an edge between nodes $n_i$, $n_j \in X$, otherwise $a_{i,j} = 0$. Typically, GAEs consist of: (1). graph encoder that takes graph $G = \{A, X\}$ as input and generates compact latent embeddings $Z$ via differential neural network layers $f(A, X; \Theta) \rightarrow Z \in \mathbb{R}^d$; (2). graph decoder that reconstructs the graph embeddings $G' = \{A', X'\}$ from latent embeddings $Z$; (3). an optimization component, with reconstruction loss $\mathcal{L}_{Rec}$, that minimizes the reconstruction error between input graph and reconstructed graph embeddings.

Our Contrastive Graph Autoencoder (CGAE) introduces a contrastive component $\mathcal{L}_C$ to GAE, minimizing the contrastive error between the compact graph embeddings $Z^G$ and the contrastive graph counterparts $Z_*^G$. Fig. 1 demonstrates the overall architecture of proposed CGAE. We first outline the novel contributions of CGAE, and introduce the graph contrastive component next.

### 2.1 Graph Representation of 2D Polygons

A polygon is defined as a simple geometry (i.e., no self-intersections) consisting of a collection of sequences of vertices encoding a single exterior ring defining the exterior boundary, and zero or multiple interior rings defining the internal holes of the geometry (Open Geospatial Consortium, 2003). The polygon geometry is converted into graph representation $G = \{A, X\}$ as input to the graph autoencoder, where $X$ captures the vertex positions and the adjacency matrix $A$ captures the adjacency of the vertices in the boundary linear ring of vertices (Fig. 1). The graph representation

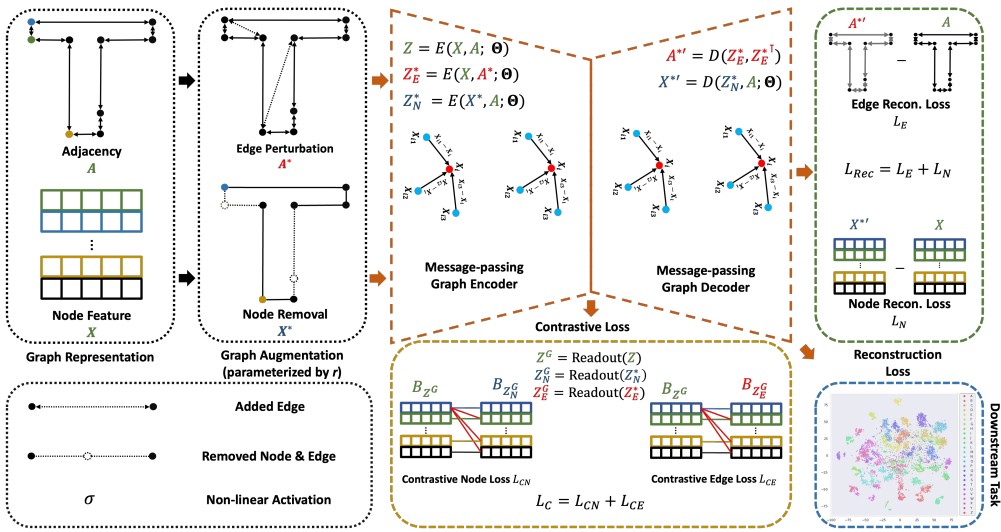

Figure 1: Model architecture of CGAE. Inputs: vertex coordinates of T-shape polygon are encoded into a node feature matrix $X$, and the connectivity of vertices into the adjacency matrix $A$. We produce contrastive pairs of $X$ and $A$ by graph augmentations, obtaining a removed node feature matrix $X^*$ and a perturbed adjacency matrix, $A^*$. CGAE learns robust and discriminative latent embeddings by the computation of the graph reconstruction loss $\mathcal{L}_{Rec}$ and contrastive loss $\mathcal{L}_C$.

enables describing the geometric and connectivity information of polygons as matrices, which are computationally efficient for graph representation learning.

## 2.2 MESSAGE-PASSING GRAPH ENCODER

Graph encoder then embeds $G$ into a latent space. Typical graph encoders use graph convolutional networks (GCN) from Kipf & Welling (2017) and its variants to learn node-wise convolution features of $X$ with the normalized graph Laplacian matrix $\hat{A} = \tilde{D}^{-1/2}\tilde{A}\tilde{D}^{-1/2}$, where $\tilde{A} = A + I$ is the adjacency matrix with an identity matrix (i.e., self-connections of nodes), and $\tilde{D}$ is the diagonal degree matrix. The latent embeddings $Z$ of graph $G$ are hence learned by a differentiable neural network layer $f(,;\Theta)$ (Eq. 1). Subsequently, we can express how GCN learns the latent node-wise embeddings $z$ of graph $G$ as per Eq. 2:

$$f(X, A; \Theta) = \tilde{D}^{-1/2}\tilde{A}\tilde{D}^{-1/2}X\Theta$$
$$= \hat{A}X\Theta. \qquad (1)$$

$$z_i = \Theta^\top \sum_{a_{i,j} \in A = 1} x'_j, \text{where } x'_j = \frac{\tilde{a}_{i,j}}{\sqrt{\tilde{d}_i \times \tilde{d}_j}}x_j. \qquad (2)$$

$\frac{\tilde{a}_{i,j}}{\sqrt{\tilde{d}_i \times \tilde{d}_j}}$ corresponds to the edge weight between nodes $x_i$ and $x_j$ of the normalized graph Laplacian matrix $\hat{A}$. We observe that a typical GCN encoder is limited to encoding polygon's boundary information and does not generalize to polygons with holes. Eq. 2 suggest that the edge weights between the node pair $(x_i, x_j)$ are constants defined by a fixed adjacency and node degree matrix, $A$ and $D$ of input graph. This largely limits the capability of graph encoders to learn expressive latent embeddings $Z$ for complex polygons with holes. Previous studies (Qi et al., 2017a;b; Wang et al., 2019) have demonstrated the importance of encoding and learning local geometric features for point sets, in particular, 3D point clouds. These learning-based models construct local $k$NN graphs from 3D point clouds and aggregate local geometric features of points in local $k$NN sub-graphs per point. These studies also show that learning local geometric features as node features of point sets extensively increases the performance of point cloud classification and segmentation. Analogically, we hypothesize that polygon boundaries can be learned as local geometric node features via

graph-based neural networks in an unsupervised graph autoencoder, leading to robust and highly expressive latent embeddings $z$ for polygon geometries with holes. We thus propose a graph-based autoencoder with a message-passing neural network as encoder (Gilmer et al., 2017) (Eq. 3):

$$z_i = \max_{a_{i,j} \in A = 1} \Theta^\top (x_i \parallel (x_j - x_i)), \tag{3}$$

where the message between nodes $x_i$ and $x_j$ is $(x_j - x_i)$, defined as the relative feature positions between the two nodes, or "node centralization" (Tailor et al., 2021). We observe that in Eq. 3 the latent embedding $z_i$ updated for node $x_i$ in a single message-passing layer encodes the maximum of the concatenation $\parallel$ of the positional feature $x_i \in X$ of a vertex and its boundary information defined by the relative positions between its neighboring nodes. A multi-layer perceptron (MLP) with trainable weights $\Theta$ linearly projects the local geometric information to a latent space and the non-linearity of the message-passing layer is then carried out by a non-linear activation function applied to the layer output $z_i$.

Compared to GCN, the message-passing layers leverage the local geometric features (i.e., relative positional features), which are previously demonstrated to be effective on 3D point sets, to learn expressive and robust node-wise embeddings for polygons. Noted that in typical graph autoencoder, the GCN layers apply the summation pooling to aggregate neighboring convolution features $x'_j$, instead in the message-passing layers, we propose to use the maximum pooling to aggregate the local geometric features. Here, the use of maximum aggregation/pooling aims to generate output features that outline representative elements and capture the skeleton of point sets (Xu et al., 2019; Qi et al., 2017a;b; Wang et al., 2019).

### 2.3 Graph Reconstruction

Graph decoder targets to reconstruct the graph data $G$ from latent embeddings $Z$, such that the reconstructed graph $G'$ to be as close as to the input graph $G$. In CGAE, we propose two kinds of decoder to reconstruct the node features $X'$ and edge features $A'$ of graph $G'$, respectively.

**Node Feature Reconstruction**  We apply a symmetric message-passing network architecture (Park et al., 2019) in CGAE's decoders to reconstruct the node feature matrix $X'$ from the latent embeddings $Z$. Concretely, given embeddings $Z$, a single-layer node decoder is defined in Eq.4 as:

$$x'_i = \max_{a_{i,j} \in A = 1} \Theta^\top (z_i \parallel (z_j - z_i)), \tag{4}$$

where the MLP eventually projects latent node-wise embedding $z_i \in Z$ to input node-wise embedding, $x'_i \in X'$. Similar to the node decoder in GAE (Yan et al., 2021), the node reconstruction loss of CGAE is based on the mean squared error (MSE) (Eq. 5):

$$\mathcal{L}_N = \frac{1}{n} \sum_{i=1}^{n} (x'_i - x_i)^2. \tag{5}$$

**Edge Reconstruction**  We add an additional edge decoder, the inner product decoder from Kipf & Welling (2016), to reconstruct the topological structure $A'$ of graph. The edge decoder leverages the latent node-wise embeddings $Z$, with $\sigma$ being a sigmoid activation function and $p(y|x)$ a conditional probability, mapping $a_{i,j} \in A'$ to $(0, 1)$ (Eq. 6):

$$A' = p(A'|Z) = \sigma(ZZ^\top), \tag{6}$$

To compute the reconstruction loss between the graph adjacency $A$ and reconstructed adjacency $A'$, we produce binary labels for edges based on the ground-truth adjacency $A$. For matrix $A$ values indexed by $\{i, j\}$, values where $a_{i,j} = 1$ are positive, and values where $a_{i,j} = 0$ are negative. The indexes of positive ground-truths are then $pos_{\{i,j\}}$ and negative ground-truths $neg_{\{i,j\}}$. For balanced binary label representation feeding to edge reconstruction, we randomly sample equal amounts of negative and positive ground-truths from adjacency $A$. We compute the edge reconstruction loss with the binary cross entropy error between the positive and negative edge labels (Eq. 7):

$$\mathcal{L}_E = -\log(A'_{pos_{\{i,j\}}}) - \log(1 - A'_{neg_{\{i,j\}}}) \tag{7}$$

where $A'_{pos_{\{i,j\}}}$ denotes the reconstructed edge value with ground-truth positive edge indexes and $A'_{neg_{\{i,j\}}}$ denotes the reconstructed edge value with ground-truth negative edge indexes.

Incorporating edge reconstruction to CGAE is necessary as the topological structure captures non-trivial connectivity information of the polygon graphs, to expressively outline the shape of the geometry. The decoders and edge reconstruction loss thus inherently force the graph encoder to learn latent embeddings that encode salient topological information of input graphs.

## 3 CONTRASTIVE GRAPH AUTOENCODER

The key contribution of CGAE is the contrastive learning of latent graph embeddings $Z^G$. At its core, contrastive learning maximizes the mutual information agreement between the positive contrastive pairs and push the negative contrastive pairs apart in training batches (Hjelm et al., 2019; Belghazi et al., 2018). It thus forces similar features to be closer in latent space.

**Graph Augmentation**    As a desirable property, if graph embeddings are robust to node corruptions (i.e., if random node elimination/addition does not drastically change the semantic representation of the polygon boundary graph (You et al., 2020)), a trained CGAE should enable reconstructing uncorrupted node features from corrupted latent embeddings $Z^*$. We apply graph-level augmentations to input data in graph $G = \{A, X\}$ and generate contrastive pairs for learning by perturbing edge information of $G$ by randomly adding an $r$ percentage of new edges to the adjacency matrix $A$, resulting in the perturbed adjacency matrix $A^*$ (Fig. 1). For node features of $G$, we randomly drop an $r$ percentage of nodes along with corresponding edges, resulting in the corrupted node feature matrix $X^*$. We define a graph augmentation function $\phi(A, X|r)$ on $G = \{A, X\}$, and $r$ as a hyper-parameter augmentation ratio. By defining $r$, $\phi(A, X|r)$ returns a corrupted node matrix $X^* \sim \phi(A, X|r)$ and a perturbed edge matrix $A^* \sim \phi(A, X|r)$, respectively.

**Corrupted Node and Perturbed Edge Reconstruction**    The message-passing graph encoder (Eq. 3) now takes a corrupted node feature $x_i^* \in X^*$ and returns a latent node-wise embedding $z_i^* \in Z_N^*$, with $Z_N^*$ representing the corrupted latent node-wise embeddings generated based on the corrupted node feature $X^*$. We then reconstruct the node-wise embeddings as proposed in Eq. 4 (applied to $z_i^*$) and compute the node reconstruction loss $\mathcal{L}_N$ as proposed in Eq. 5 (applied to $x_i^{*\prime}$).

Similarly, the principal prior of edge perturbation is that randomly adding edges as noise to graph connectivity should not alter its semantic information. We therefore reconstruct $G$'s connectivity from latent embeddings $Z^*$ with perturbed edge information. Correspondingly, the message-passing graph encoder in Eq. 3 takes perturbed edge connection $a_{i,j}^* \in A^*$ as inputs and returns a latent node-wise embedding $z_i^* \in Z_E^*$, where $Z_E^*$ represents the latent node-wise embeddings generated based on the perturbed edge matrix $A^*$.

We then reconstruct the graph connectivity $A^{*\prime}$ by applying Eq. 6 and compute the edge reconstruction loss $\mathcal{L}_E$ as proposed in Eq. 7, applied on $A^{*\prime}_{pos_{\{i,j\}}}$ and $A^{*\prime}_{neg_{\{i,j\}}}$. The resulting latent embeddings should thus be less sensitive to arbitrary variations of boundary edges of polygon geometries, thus robustly capturing significant shape and semantic information.

**Contrastive Learning**    The main objective of contrastive learning with graph autoencoders is to maximize the mutual information between the global latent graph embeddings and their augmented counterparts. Graph encoders thus learn to encode compact graph and node-wise embeddings tolerant to node corruptions (i.e., random node dropping) and edge perturbation (i.e., random edge addition). The latent embedding representing polygon shapes thus become more robust to geometric transformations and minor variations of boundary information.

We thus compute three kinds of latent graph embeddings $Z^G$, $Z_N^G$ and $Z_E^G \in \mathbb{R}^d$ from the latent node-wise embeddings $Z$, $Z_N^*$ and $Z_E^* \in \mathbb{R}^{n \times d}$ with a graph readout function: $Z^G = Readout_{mean}(Z)$, where $Readout_{mean}(\cdot)$ is a global average pooling to compute graph-wise representations from node-wise embeddings. We put latent graph embeddings in training batches with pre-defined batch size $b$ ($b = 32$), giving: $B_{Z^G} = [Z_1^G, Z_2^G, ..., Z_b^G]^\top$, $B_{Z_N^G} = [Z_{N_1}^G, Z_{N_2}^G, ..., Z_{N_b}^G]^\top$, $B_{Z_E^G} = [Z_{E_1}^G, Z_{E_2}^G, ..., Z_{E_i}^G]^\top$, where $B_{Z^G}$, $B_{Z_N^G}$ and $B_{Z_E^G} \in \mathbb{R}^{b \times d}$ are training batches of latent

graph embeddings $Z^G$, $Z^G_N$ and $Z^G_E$. A contrastive loss function is then defined to maximize the mutual information agreement of positive contrastive pairs and push negative contrastive pairs apart in batches. Here, we apply the normalized temperature-scaled cross-entropy loss (NT-Xent) from Sohn (2016) and Chen et al. (2020) to compute the contrastive loss of contrastive pairs $L_{CN}(Z^G_i, Z^G_{N_i})$ and $L_{CE}(Z^G_i, Z^G_{E_i})$ as in Eq. 8:

$$\mathcal{L}_{CN}(Z^G_i, Z^G_{N_i}) = -\log \frac{\exp(sim(Z^G_i, Z^G_{N_i})/\tau)}{\sum_i^b \sum_{j,i\neq j}^b \exp(sim(Z^G_i, Z^G_{N_j})/\tau)}, \tag{8}$$

where $\tau$ is a temperature-scaled value that controls the impact of positive (i.e., $(Z^G_i, Z^G_{N_i})$) vs. negative (i.e., $(Z^G_i, Z^G_{N_j})$) contrastive pairs, and $sim(\cdot)$ is a similarity function that measures the similarity of embeddings, which is defined as the cosine similarity. Following Eq. 8, the average NT-Xent loss on batches $(B_{Z^G}, B_{Z^G_N})$ and $(B_{Z^G}, B_{Z^G_E})$ is defined as (Eq. 9):

$$\mathcal{L}_{CN}(B_{Z^G}, B_{Z^G_N}) = \frac{1}{b}\sum_{i=1}^{b} \mathcal{L}_{CN}(Z^G_i, Z^G_{N_i}). \tag{9}$$

We optimize the CGAE with the reconstruction loss $\mathcal{L}_{Rec}$ and contrastive loss $\mathcal{L}_C$. According to Eq. 5 and 7, $\mathcal{L}_{Rec}$ is defined as $\mathcal{L}_{Rec} = \mathcal{L}_N + \mathcal{L}_E$, and based on Eq. 8 and Eq. 9, $\mathcal{L}_C$ is defined as $\mathcal{L}_C = \mathcal{L}_{CN} + \mathcal{L}_{CE}$. We then optimize the CGAE by minimizing the sum of losses $\mathcal{L} = \mathcal{L}_{Rec} + \mathcal{L}_C$. CGAE is designed to cluster graph embeddings of complex shapes using latent node-wise embeddings $Z$ generated from the message-passing encoder in CGAE. The global graph embeddings $Z^G$ are pooled by a $readout$ layer (i.e., $Readout_{mean}$) from $Z$. Since graph embeddings $Z^G$ encode the geometric and boundary information of polygons in latent space, polygons with geometrically similar shapes are clustered, and hence retrieved by finding the $k$ nearest neighbours ($k$NN) of latent graph embeddings of query shapes.

## 4 EXPERIMENTS

We evaluate the performances of proposed CGAE against two baselines: GAE (Yan et al., 2021) and NUFT (Mai et al., 2023; Jiang et al., 2019) on polygon retrieval from the Glyph, Open Street Map (OSM) and Melbourne datasets (Appendix A). We conduct an ablation study to demonstrate the effectiveness of the proposed message-passing encoder-decoder structure and the contrastive components of CGAE. Stepwise, we first replace the GCN backbone in GAE with a message-passing backbone, and enrich the baseline GAE (i.e., GAE ($\mathcal{L}_N$)) by adding the edge reconstruction loss $\mathcal{L}_E$ (i.e., GAE ($\mathcal{L}_N + \mathcal{L}_E$)), and the proposed contrastive loss $\mathcal{L}_C$, resulting in CGAE and evaluate performance gains. The progressively refined GAEs are applied on three distinct backbone networks: (1). the GCN by Kipf & Welling (2017), (2). the message-passing neural network: EdgeConv (Wang et al., 2019), to test the benefit of learning local geometric features (i.e., the relative positions between nodes), and (3). the graph isomorphic network (GIN (Xu et al., 2019)). Model training and parameters for experiments are described in Appendix B.

**Model Discriminability (Glyph Dataset)**   A robust and effective autoencoder generates highly discriminative latent embeddings for geometrically transformed polygons. We train a CGAE (Edge-Conv backbone) and a baseline GAE on the synthetic Glyph-O dataset, and visualize the respective latent embeddings in 2D space via a t-SNE plot (Van der Maaten & Hinton, 2008) (Fig. 2). The latent embeddings of CGAE are highly discriminative and less noisy than in GAE. This qualitatively indicates the effectiveness of incorporating graph augmentation, contrastive learning, and message-passing layers into graph autoencoder architectures. To investigate the gains brought by these components individually, we quantitatively evaluate the performance on a retrieval task with geometrically transformed (rotated, skewed and scaled) shapes (Glyph dataset).

Table 1 reports the polygon retrieval performance of GAEs on the Glyph dataset using shape similarity (Hausdorff distance, Appendix D) between the query shape and its $k$ nearest neighbors (for $k$ 1 to 6). The CGAE-EdgeConv achieves the best performance on Glyph-O, Glyph-R, Glyph-SC and Glyph-SK datasets for all $k$. By ablation we identify the performance gains of CGAE brought by edge reconstruction loss $\mathcal{L}_E$ over the baseline GAE. Reconstructing the edge connectivity generates latent embeddings that encode information enabling the effective discovery of similar shapes.

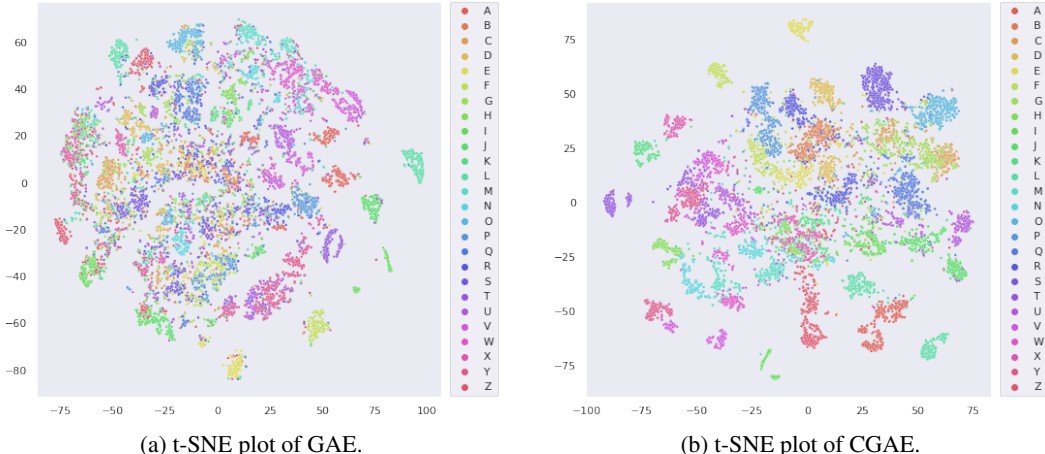

(a) t-SNE plot of GAE.                    (b) t-SNE plot of CGAE.

Figure 2: t-SNE plot of latent feature embeddings (auto learning rate, perplexity=50, iterations=1000) of the feature embeddings of models tested on dataset Glyph-O.

By comparing models with different backbones, EdgeConv achieves the best performance on both GAE and CGAE. This suggests that message-passing graph encoders learning of local geometric information (i.e., node centralization and relative positions) is key for superior performance on shape retrieval w.r.t geometric transformations. While the CGAE performs best on all Glyph datasets, contrastive learning has the highest impact on encoding skewed polygons, where in Glyph-SK CGAE-EdgeConv leads to largest improvement compared to GAE-EdgeConv.

**Building Polygon Retrieval (OSM Dataset)**   Table 2 reports the polygon retrieval performance on two variants of the OSM building dataset: (1) original polygons (OSM-O) and (2) rotated and mirrored polygons (OSM-R). The CGAE performs best among all models on both. By ablation, we find that GAE with edge reconstruction loss $\mathcal{L}_E$ generalizes better on both OSM-O and OSM-R building footprints compared to baseline GAE but with limited performance gains (OSM-O: 1-NN $(0.218 \rightarrow 0.208)$ to 6-NN $(0.271 \rightarrow 0.267)$ and OSM-R: 1-NN $(0.323 \rightarrow 0.308)$ to 6-NN $(0.400 \rightarrow 0.396)$). Message-passing backbone networks (either GIN or EdgeConv) bring significant model generalizability (Table 2). Extracting shapes from geometrically transformed datasets is challenging, yet CGAE with contrastive loss and edge reconstruction loss $\mathcal{L}_E + \mathcal{L}_C$ is robust to polygon rotations and reflections. Typically, CGAE-EdgeConv shows large improvements on OSM-R compared with GAE-EdgeConv (i.e., 1-NN $(0.235 \rightarrow 0.225)$ to 6-NN $(0.320 \rightarrow 0.301)$).

We emphasize that all models were only trained on the synthetic Glyph dataset, and evaluated on the OSM building dataset to explore model generalizability. Compare the performance on Glyph-O and Glyph-R (Table 1) with that on OSM-O and OSM-R (Table 2). The baseline GCN achieves comparable performances on both Glyph-O and OSM-O datasets (from 1-NN $(0.193 \rightarrow 0.218)$ to 6-NN $(0.271 \rightarrow 0.271)$) but the model degrades when generalizing from Glyph-R to OSM-R: 1-NN $(0.271 \rightarrow 0.323)$ to 6-NN $(0.343 \rightarrow 0.400)$. In contrast, the CGAE-EdgeConv only suffers limited degradation generalizing from Glyph-R to OSM-R: from 1-NN $(0.212 \rightarrow 0.225)$ to 6-NN $(0.269 \rightarrow 0.301)$.

**Building Polygon Retrieval (Melbourne Dataset)**   We finally test the large-scale real-world shape retrieval performance of GAEs on the Melbourne building dataset, with uncontrolled shapes, where GAEs were trained on the synthetic Glyph dataset. Table 3 shows that CGAE-EdgeConv again outperforms other models by a large margin, especially the baseline GAE that cannot handle multipart polygons. Comparing CGAE-EdgeConv with GAE-EdgeConv implies that the incorporation of contrastive loss brings positive effects to retrieval of multipart (incl. with holes) polygons (from 1-NN $(0.223 \rightarrow 0.214)$ to 6-NN $(0.297 \rightarrow 0.278)$). Overall, the proposed CGAE demonstrates a strong capability of latent feature encoding even for complex shapes. The model can identify and retrieve similar shapes, even from large-scale real-world geometry datasets and on shapes with nuanced geometries of the kind never seen in training.

Table 1: CGAE performance against baselines on the Glyph dataset. Hausdorff distance – best in bold (the smaller the better).

| Dataset | Model | Backbone | 1-NN | 2-NN | 3-NN | 4-NN | 5-NN | 6-NN |
|---------|-------|----------|------|------|------|------|------|------|
| Glyph-O | NUFT | MLP | 0.304 | 0.338 | 0.354 | 0.365 | 0.371 | 0.379 |
| | GAE $(\mathcal{L}_N)$ | GCN | 0.193 | 0.229 | 0.245 | 0.257 | 0.264 | 0.271 |
| | GAE $(\mathcal{L}_N + \mathcal{L}_E)$ | GCN | 0.183 | 0.218 | 0.236 | 0.245 | 0.252 | 0.260 |
| | | GIN | 0.159 | 0.192 | 0.207 | 0.217 | 0.224 | 0.232 |
| | | EdgeConv | 0.152 | 0.183 | 0.199 | 0.211 | 0.218 | 0.225 |
| | CGAE $(\mathcal{L}_N + \mathcal{L}_E + \mathcal{L}_C)$ | GCN | 0.174 | 0.207 | 0.222 | 0.232 | 0.239 | 0.245 |
| | | GIN | 0.159 | 0.189 | 0.203 | 0.213 | 0.219 | 0.226 |
| | | EdgeConv | **0.150** | **0.180** | **0.193** | **0.203** | **0.209** | **0.213** |
| Glyph-R | NUFT | MLP | 0.417 | 0.442 | 0.456 | 0.463 | 0.470 | 0.472 |
| | GAE $(\mathcal{L}_N)$ | GCN | 0.271 | 0.302 | 0.318 | 0.328 | 0.335 | 0.343 |
| | GAE $(\mathcal{L}_N + \mathcal{L}_E)$ | GCN | 0.263 | 0.295 | 0.311 | 0.321 | 0.329 | 0.340 |
| | | GIN | 0.231 | 0.261 | 0.275 | 0.287 | 0.296 | 0.304 |
| | | EdgeConv | 0.219 | 0.248 | 0.263 | 0.275 | 0.285 | 0.292 |
| | CGAE $(\mathcal{L}_N + \mathcal{L}_E + \mathcal{L}_C)$ | GCN | 0.245 | 0.270 | 0.285 | 0.294 | 0.304 | 0.311 |
| | | GIN | 0.222 | 0.244 | 0.257 | 0.266 | 0.272 | 0.278 |
| | | EdgeConv | **0.212** | **0.235** | **0.246** | **0.256** | **0.262** | **0.269** |
| Glyph-SK | NUFT | MLP | 0.465 | 0.488 | 0.495 | 0.501 | 0.502 | 0.505 |
| | GAE $(\mathcal{L}_N)$ | GCN | 0.284 | 0.310 | 0.321 | 0.330 | 0.337 | 0.344 |
| | GAE $(\mathcal{L}_N + \mathcal{L}_E)$ | GCN | 0.272 | 0.301 | 0.313 | 0.321 | 0.330 | 0.337 |
| | | GIN | 0.246 | 0.278 | 0.294 | 0.306 | 0.314 | 0.322 |
| | | EdgeConv | 0.234 | 0.261 | 0.278 | 0.290 | 0.300 | 0.307 |
| | CGAE $(\mathcal{L}_N + \mathcal{L}_E + \mathcal{L}_C)$ | GCN | 0.256 | 0.279 | 0.291 | 0.300 | 0.307 | 0.312 |
| | | GIN | 0.228 | 0.249 | 0.260 | 0.268 | 0.272 | 0.278 |
| | | EdgeConv | **0.219** | **0.240** | **0.252** | **0.261** | **0.266** | **0.271** |
| Glyph-SC | NUFT | MLP | 0.336 | 0.360 | 0.374 | 0.379 | 0.390 | 0.393 |
| | GAE $(\mathcal{L}_N)$ | GCN | 0.229 | 0.257 | 0.268 | 0.278 | 0.287 | 0.293 |
| | GAE $(\mathcal{L}_N + \mathcal{L}_E)$ | GCN | 0.224 | 0.253 | 0.265 | 0.279 | 0.288 | 0.293 |
| | | GIN | 0.189 | 0.215 | 0.230 | 0.240 | 0.248 | 0.257 |
| | | EdgeConv | 0.181 | 0.205 | 0.221 | 0.230 | 0.239 | 0.245 |
| | CGAE $(\mathcal{L}_N + \mathcal{L}_E + \mathcal{L}_C)$ | GCN | 0.208 | 0.230 | 0.243 | 0.252 | 0.260 | 0.262 |
| | | GIN | 0.182 | 0.203 | 0.213 | 0.219 | 0.225 | 0.230 |
| | | EdgeConv | **0.173** | **0.194** | **0.205** | **0.213** | **0.219** | **0.223** |

Table 2: CGAE performance against baselines on the OSM building dataset. Hausdorff distance – best in bold (the smaller the better). Qualitative results are displayed in Appendix E.1.

| Dataset | Model | Backbone | 1-NN | 2-NN | 3-NN | 4-NN | 5-NN | 6-NN |
|---------|-------|----------|------|------|------|------|------|------|
| OSM-O | NUFT | MLP | 0.302 | 0.327 | 0.341 | 0.352 | 0.354 | 0.363 |
| | GAE $(\mathcal{L}_N)$ | GCN | 0.218 | 0.238 | 0.250 | 0.261 | 0.265 | 0.271 |
| | GAE $(\mathcal{L}_N + \mathcal{L}_E)$ | GCN | 0.208 | 0.232 | 0.245 | 0.254 | 0.259 | 0.267 |
| | | GIN | 0.187 | 0.209 | 0.221 | 0.229 | 0.237 | 0.243 |
| | | EdgeConv | 0.174 | 0.191 | 0.202 | 0.212 | 0.219 | 0.228 |
| | CGAE $(\mathcal{L}_N + \mathcal{L}_E + \mathcal{L}_C)$ | GCN | 0.200 | 0.220 | 0.231 | 0.238 | 0.244 | 0.251 |
| | | GIN | 0.184 | 0.203 | 0.213 | 0.221 | 0.227 | 0.231 |
| | | EdgeConv | **0.172** | **0.190** | **0.200** | **0.208** | **0.214** | **0.217** |
| OSM-R | NUFT | MLP | 0.431 | 0.461 | 0.481 | 0.490 | 0.502 | 0.508 |
| | GAE $(\mathcal{L}_N)$ | GCN | 0.323 | 0.353 | 0.373 | 0.384 | 0.394 | 0.400 |
| | GAE $(\mathcal{L}_N + \mathcal{L}_E)$ | GCN | 0.308 | 0.344 | 0.359 | 0.379 | 0.387 | 0.396 |
| | | GIN | 0.267 | 0.302 | 0.325 | 0.340 | 0.353 | 0.362 |
| | | EdgeConv | 0.235 | 0.266 | 0.285 | 0.302 | 0.311 | 0.320 |
| | CGAE $(\mathcal{L}_N + \mathcal{L}_E + \mathcal{L}_C)$ | GCN | 0.292 | 0.320 | 0.338 | 0.351 | 0.359 | 0.369 |
| | | GIN | 0.245 | 0.277 | 0.296 | 0.310 | 0.317 | 0.326 |
| | | EdgeConv | **0.225** | **0.255** | **0.272** | **0.286** | **0.295** | **0.301** |

Table 3: CGAE performance against baselines on the Melbourne building dataset. Hausdorff distance – best in bold (the smaller the better). Qualitative results are displayed in Appendix E.2.

| Dataset | Model | Backbone | 1-NN | 2-NN | 3-NN | 4-NN | 5-NN | 6-NN |
|---------|-------|----------|------|------|------|------|------|------|
| Melbourne | NUFT | MLP | 0.382 | 0.401 | 0.413 | 0.418 | 0.424 | 0.429 |
| | GAE ($\mathcal{L}_N$) | GCN | 0.588 | 0.597 | 0.598 | 0.604 | 0.602 | 0.608 |
| | GAE ($\mathcal{L}_N + \mathcal{L}_E$) | GCN | 0.579 | 0.588 | 0.591 | 0.592 | 0.596 | 0.598 |
| | | GIN | 0.566 | 0.572 | 0.577 | 0.579 | 0.581 | 0.583 |
| | | EdgeConv | 0.223 | 0.255 | 0.273 | 0.283 | 0.290 | 0.297 |
| | CGAE ($\mathcal{L}_N + \mathcal{L}_E + \mathcal{L}_C$) | GCN | 0.575 | 0.584 | 0.587 | 0.590 | 0.589 | 0.593 |
| | | GIN | 0.555 | 0.562 | 0.565 | 0.568 | 0.570 | 0.572 |
| | | EdgeConv | **0.214** | **0.242** | **0.257** | **0.266** | **0.273** | **0.278** |

## 5 DISCUSSION

Node centralization in message-passing graph autoencoders is key for shape retrieval performance as shown through ablation studies on synthetic (Glyph), controlled (OSM), and uncontrolled (Melbourne) datasets, compared to graph (Laplacian) convolution layers. The relative node position message computed in EdgeConv (i.e., node centralization) and passed between nodes of local sub-graphs encodes highly expressive local geometric information. While this has been previously shown effective on unstructured 3D point cloud learning (Qi et al., 2017a;b; Wang et al., 2019), this study documents the significance of learning local geometric features on 2D vector polygons.

Joint reconstruction of edge and node features from latent node-wise embeddings enables GAEs to exploit the structural and geometric information for finer feature encoding. Our ablation study shows that the GAE with the reconstruction loss $\mathcal{L}_N + \mathcal{L}_E$ overperforms baseline GAE with $\mathcal{L}_N$ independently of network backbones. This indicates the importance of learning graph connectivity and structural information for graph feature encoding.

Graph augmentation and contrastive learning further improve latent feature generalization and robustness as shown by CGAE's performance on Glyph-R, Glyh-SK (Table 1) and OSM-R (Table 2). The CGAE-EdgeConv combination achieves best performance on geometrically transformed polygons over baseline GAE-EdgeConv. The information captured by the latent graph embedding is enhanced through contrastive learning and contrastive loss from augmented signals (i.e., based on random node dropping and edge perturbation). This supports our hypothesis of graph encoders' ability to learn compact graph embeddings that are tolerant to node corruptions and edge perturbations.

CGAE's superior performance (Table 3) demonstrates how contrastive learning aids in model generalisation. The polygons in Glyph and OSM datasets have comparatively simple boundaries (i.e., small vertex counts) that encapsulate distinct geometric and semantic information (i.e., shapes of characters), whereas the boundaries of complex building footprints in Melbourne dataset may consist of many vertices. CGAE generalizes effectively the geometric information learned from simple polygons to complex shapes, demonstrating a desirable model property, i.e., decoupling of shape detail (i.e., polygon trivial vertex count) from retrieval accuracy.

## 6 CONCLUSION

We present a novel unsupervised contrastive graph autoencoder (CGAE) for robust polygon retrieval of building polygons. We conduct ablation studies on multiple polygonal datasets to demonstrate the effectiveness of proposed CGAE, and present empirical evidence that the proposed CGAE with graph message-passing encoder-decoder, multiple reconstruction losses and contrastive learning outperforms a baseline graph autoencoder with a single node reconstruction loss, as well as all baselines, even without loss weight tunning. Our proposed CGAE shows desirable characteristics for retrieval of building polygons: is **robust to variable geometry vertex counts**; can retrieve **complex polygons with or without holes**; is robust to polygon **reflections and rotations**; can effectively **generalize** to large-scale polygon maps. The generalisation of CGAE to arbitrary graphs that only capture topology ( e.g., social networks) but not graph geometry or the application to ensembles of shapes defining semantics (terrace houses)Lüscher et al. (2009) are not addressed here.

## REPRODUCIBILITY STATEMENT

Source codes for implementing models and reproducing experiment results is available in anonymous link: https://figshare.com/s/47052e1dde02ca506085.

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

APPENDIX

The appendix is organized into the following sections:

- Appendix A Details of Glyph polygons, OSM and Melbourne building footprint datasets
- Appendix B Model parameters & training setup
- Appendix C Listing of notations and learnable parameters in ML models
- Appendix D Details of Hausdorff distance metric
- Appendix E Qualitative results that support results in Section 4
- Appendix F Qualitative results of Non-ML based polygon shape retrieval methods
- Appendix G Persistent diagrams describing the topological features of query shapes tested in Melbourne dataset

## A  DATASETS

To evaluate the performance of CGAE, we conduct retrieval experiments on three polygon datasets and compare model performance with the baseline model GAE of Yan et al. (2021). For consistent shape performance on the three 2D polygon datasets noted below, we center and normalize the coordinates of polygon vertices to the range $[-1, 1]$.

**Glyph polygons**. Anonymous (2023) introduce a synthetic dataset of highly variable geometric shapes for model training and evaluation. The synthetic Glyph dataset consists of 26 Latin alphabet glyph geometries of alphabet letters (semantic class $A$ to $Z$) of fonts gathered from an online source (Google, 2010). Anonymous (2023) extract the boundaries of glyphs for 1,413 sans serif and 1,002 serif fonts, to produce 2D simple polygon geometries compliant with Open Geospatial Consortium (2003). Geometric transformations (i.e., rotate, scale and skew) are applied to 2D simple polygons for data augmentations. Each glyph geometry sample was rotated by a random angle in $[-75°, 75°]$; sheared by a random angle in $[-45°, 45°]$ on the $x$ and $y$ axes; and scaled by a random factor in $[0.1, 2]$ on the $x$ and $y$ axes. After geometric transformations, four categories of datasets are generated, including the original dataset: Glyph-O for original polygons, Glyph-R for rotated polygons, Glyph-SC for scaled polygons, and Glyph-SK for skewed polygons. The four Glyph datasets are combined and divided into a training/validation/test set $(60 : 20 : 20)$. Examples of glyph geometries are displayed in Fig. 3.

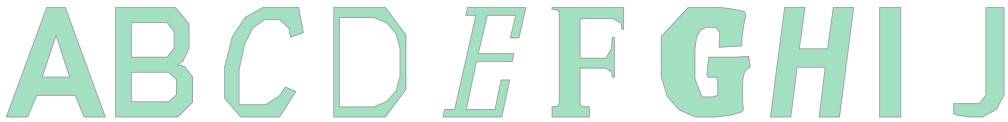

Figure 3: The top 10 samples from Glyph-O dataset in semantic class $A$ to $J$.

**OSM buildings**. We further use the benchmark dataset by Yan et al. (2021) for mode evaluation. The dataset contains 10,000 real-world building footprints extracted from OSM (OpenStreetMap contributors, 2023), labelled in 10 categories based on a template matching to letters (Yan et al., 2017). $50\%$ of building footprints are randomly rotated or reflected. We categorize the dataset into: OSM-O and OSM-R, where OSM-O contains 5000 canonical building footprints, and OSM-R contains 5000 randomly rotated or reflected buildings. The dataset includes additional standard polygons as query shapes, shown in Fig. 4. With evaluate how the performance of CGAE trained on a synthetic polygonal dataset generalizes to a real-world building footprint dataset, compared to the GAE baseline.

**Melbourne footprints**. While the synthetic Glyph dataset includes polygon geometries with holes (i.e., Latin alphabet letters with holes) and dataset OSM includes real-world building footprints without holes, we further explore the capability of CGAE in encoding information of polygons with extremely variable shapes and numerous holes for polygon retrieval. We use the open Melbourne building footprints dataset for evaluation (CoM Open Data, 2021), comprising the footprints of all structures within the City of Melbourne in Australia (Fig.5).

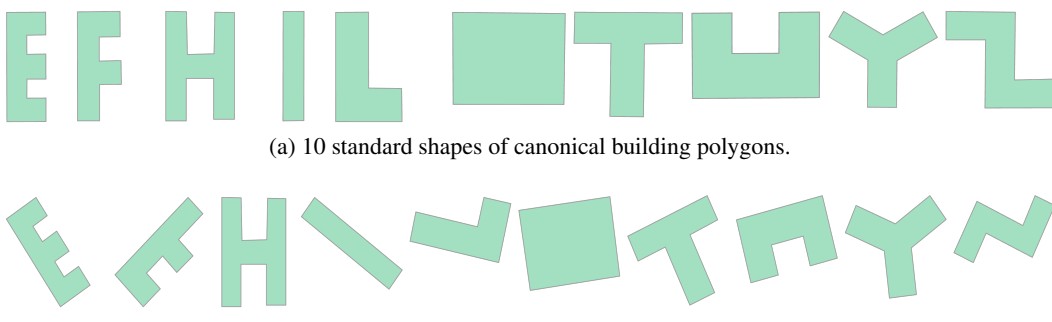

(a) 10 standard shapes of canonical building polygons.

(b) 10 standard shapes of rotated or reflected building polygons.

Figure 4: Standard shapes of dataset OSM in 10 categories: E, F, H, I, L O, T, U, Y, Z-shape polygons.

While these geometries are annotated with semantic labels such as Building structure, Tram stop, Bridge, Jetty, Toilet, Train Platform or Ramp. We only investigate models' performance on retrieving building structures of similar shapes and exclude these semantic categories, as they do not directly map to footprint shape.

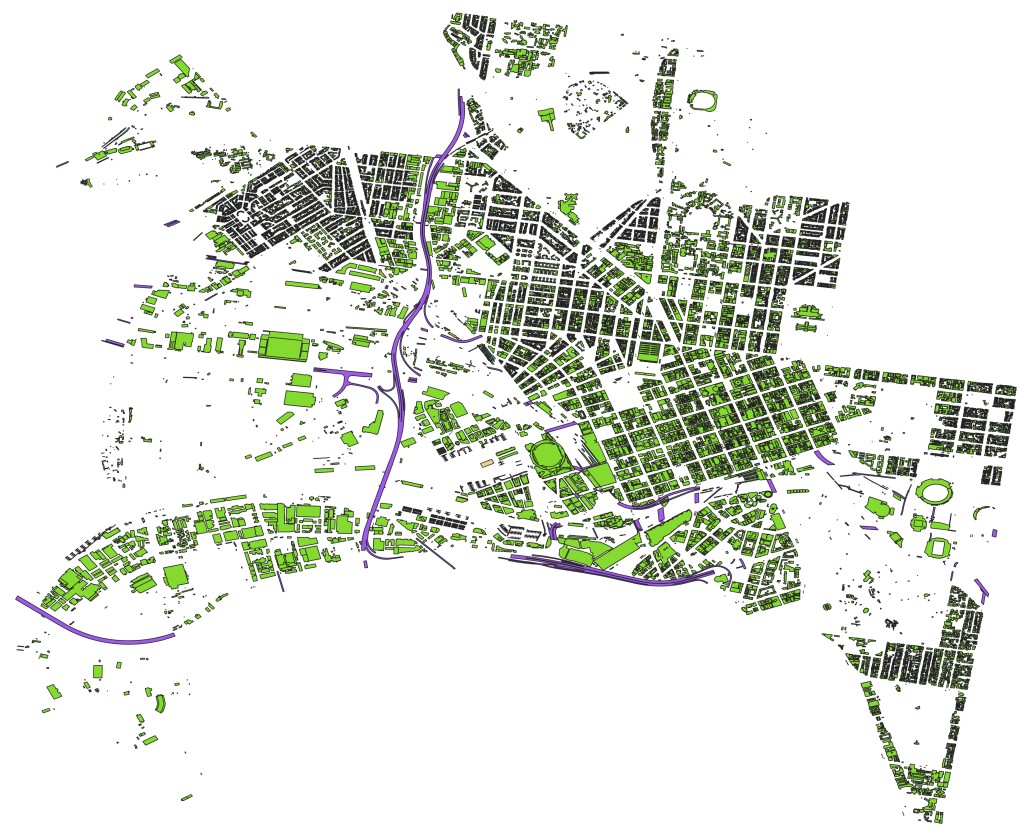

Figure 5: The map visualisation of Melbourne building footprint dataset (CoM Open Data, 2021), recorded in May, 2020. Purple structures are bridges and green structures are buildings.

## B  MODEL TRAINING & PARAMETERS

We train all models for 100 epochs with the Adam optimizer (Kingma & Ba, 2015) and an initial learning rate of 0.0001. We reduce the scale of the learning rate with a cosine annealing schedule (Loshchilov & Hutter, 2017) for better model convergence. We set the training batch size $b = 32$ and apply the same batch size to contrastive loss in CGAE. We set the augmentation ratio $r$ to $20\%$ for both random node dropping and edge perturbation in graph augmentation.

## C  ALGORITHM STATEMENT

We depict the algorithmic sequence of CGAE and its relationship with the Equations noted in the main paper in Fig. 6.

Figure 6: Algorithm statement of CGAE, which includes a listing of learnable parameters and references to equations Eq. 3 - 8.

## D  EVALUATION METRIC

We use the Hausdorff distance metric (Rucklidge, 1996) to measure the similarity between the query and extracted polygons. The Hausdorff distance has the advantage of taking the position, shape and orientation of objects measured into account when computing the similarity of two geometries (i.e., point sets) (Veltkamp, 2001; Min et al., 2007). Let A and B represent two closed point sets, and $p_a$, $p_b$ are two points $\in$ A and B, respectively. $\| \cdot \|$ is a distance metric, typically Euclidean distance. The Hausdorff distance $H(A, B) \in [0.0, +\infty]$ is the maximum of all the shortest distances from $p_a \in A$ to the closest point $p_b \in B$, or formally (Eq. 10):

$$
\begin{aligned}
H(A, B) &= \max\{h(A, B), h(B, A)\}, \; where \\
h(A, B) &= \max_{p_a \in A}\{ \min_{p_a \in B} \|p_a - p_b\| \} \\
h(B, A) &= \max_{p_b \in B}\{ \min_{p_A \in A} \|p_a - p_b\| \}.
\end{aligned}
\tag{10}
$$

# E  QUALITATIVE RESULTS

## E.1  OSM DATASET

We present the polygon retrieval performance of GAEs with ten standard polygons (Fig. 4) as queries, on the OSM dataset (Yan et al., 2021), illustrating the qualitative performances of baseline GAE and CAGE on the OSM dataset in Fig. 7 and 8 (OSM-O), Fig. 10 and 11 (OSM-R), respectively. We also add the results of the benchmark NUFT method in Fig. 9 (OSM-O) and Fig. 12 (OSM-R).

CGAE handles both simple query shapes as well as more challenging query templates when retrieving geometries from OSM-R, matching semantic categories and shapes. In contrast, the baseline GAE only extracts visually similar polygons with correct semantic categories for simple query shapes, such as Y, U, I, O and H, and fails to differentiate shapes with added boundary complexity, e.g., E, F and T (Fig. 10). Baseline GAE incorrectly extracts multiple I and L-shaped polygons as matches for the query template E, and U and I-shaped polygons for a F query shape, as reflected in the high Hausdorff distance indicating shape dissimilarity between the extracted and query polygons.

The proposed CGAE performs more robustly on the rotated query shapes and extracts polygons with the correct semantic category for E (Fig. 11). For the challenging F-shaped query, CGAE deteriorates only from the 3rd position (with an H-shaped polygon) and on the 5th and 6-NN (extracting E-shaped polygons). Note that overall, the incorrectly retrieved polygons are still geometrically more similar to the query shape, as reflected by the low Hausdorff distance.

This further illustrates the desired qualities of the proposed model CGAE, i.e., robustness to polygon rotation and reflection, and the ability to generalize to real-world shapes (e.g., building footprints from OSM).

We note that comparatively to GAE and CGAE, the NUFT-based method struggles, in the case of the shape of L already at the 1-NN on OSM-O, and produces somewhat unrealiable results on OSM-R (e.g., E).

## E.2  MELBOURNE DATASET

We present the polygon retrieval performance of GAEs on the Melbourne building footprint dataset qualitatively in Fig. 13 and 15 (Melbourne), Fig. 14 and 16 (Melbourne simplified), and add the results of the benchmark NUFT method in Fig. 17 (Melbourne) and Fig. 18 (Melbourne simplified).

For simpler circular query polygons without internal holes, CGAE successfully retrieves geometrically similar polygons for query geometries with low Hausdorff distances ($< 0.5$) while GAE identifies shapes with a relatively higher Hausdorff distance ($> 0.5$). With more complex circular polygons with a single hole, both GAE and CGAE successfully identify matching counterparts. CGAE finds highly similar geometries with complex exteriors, while GAE returns polygons with over-simplified exteriors, less geometrically similar to the query templates. This performance of GAE is deteriorating with more and more complex shapes, while CGAE maintains robust performance.

Further, to qualitatively verify that the CGAE is independent of polygonal vertex counts, we present the retrieval results with query shapes simplified by Douglas & Peucker (1973) for Melbourne dataset.

We note that compared to GAE and CGAE, the NUFT-based method underperforms. In particular, we note a deterioration of performance when it comes to polygons with holes, where this significant characteristic of the shapes is, inconsistently, neglected.

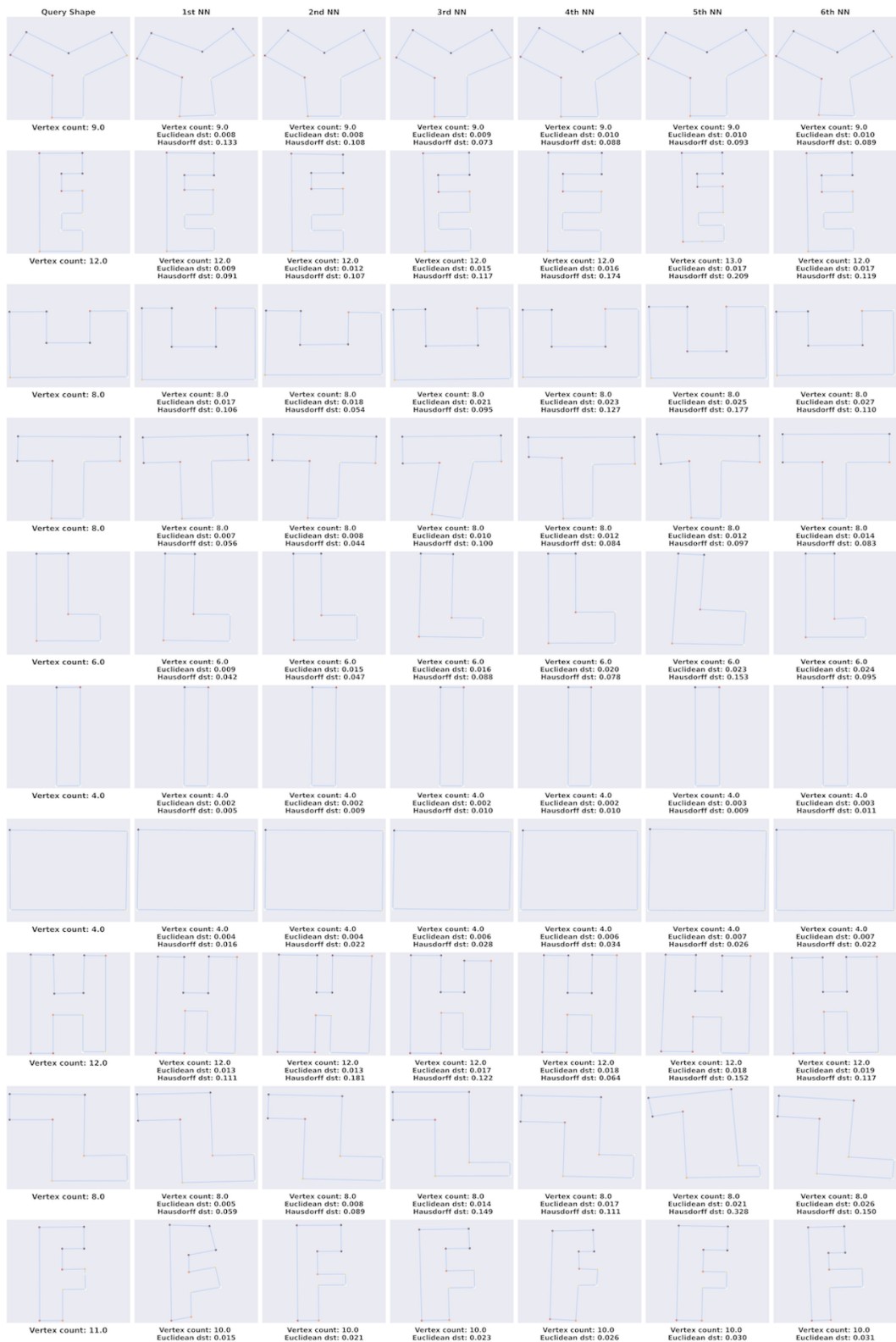

Figure 7: Polygon retrievals of GAE-GCN (baseline) on OSM-O using ten standard shapes as queries.

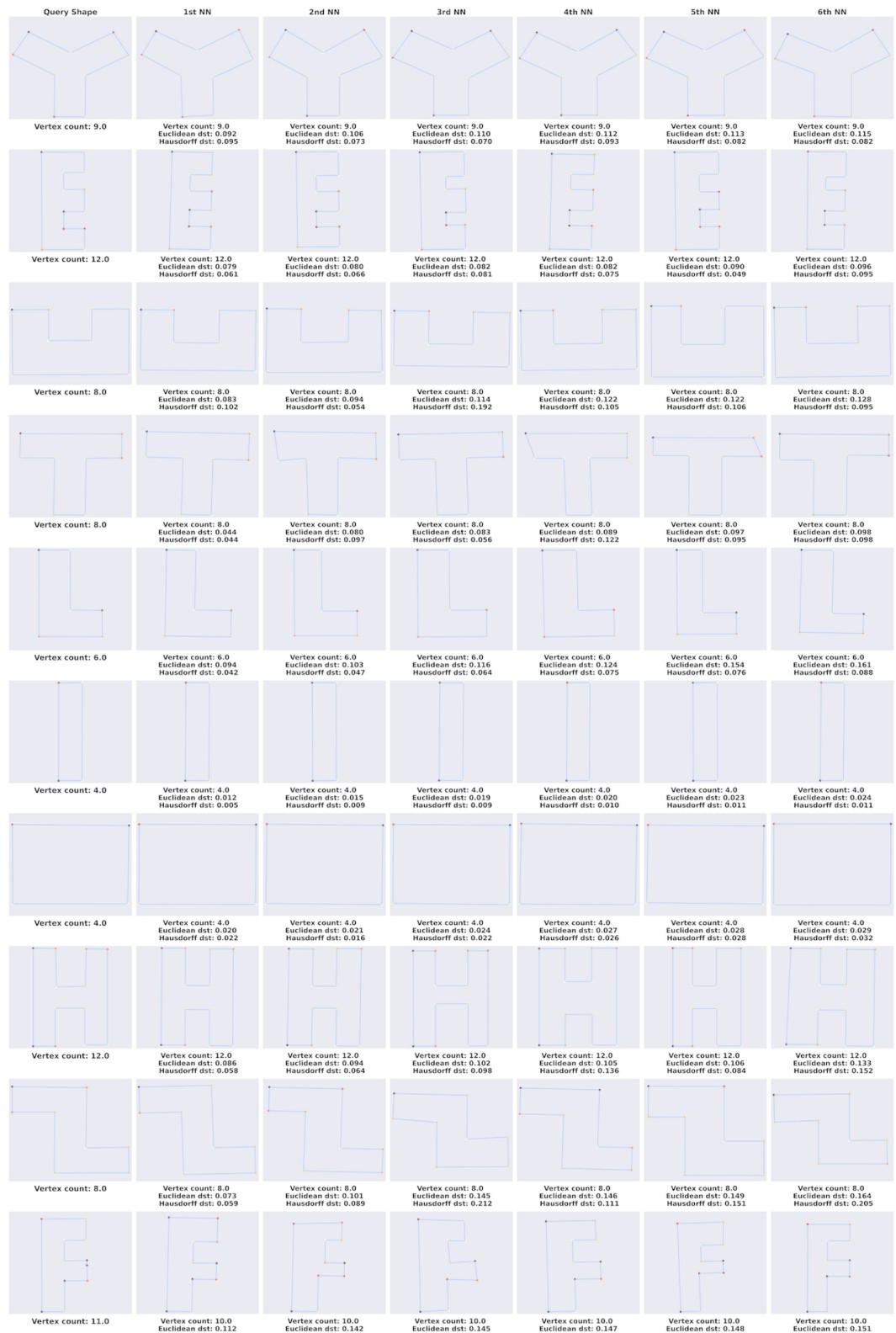

Figure 8: Polygon retrievals of CGAE-EdgeConv on OSM-O using ten standard shapes as queries.

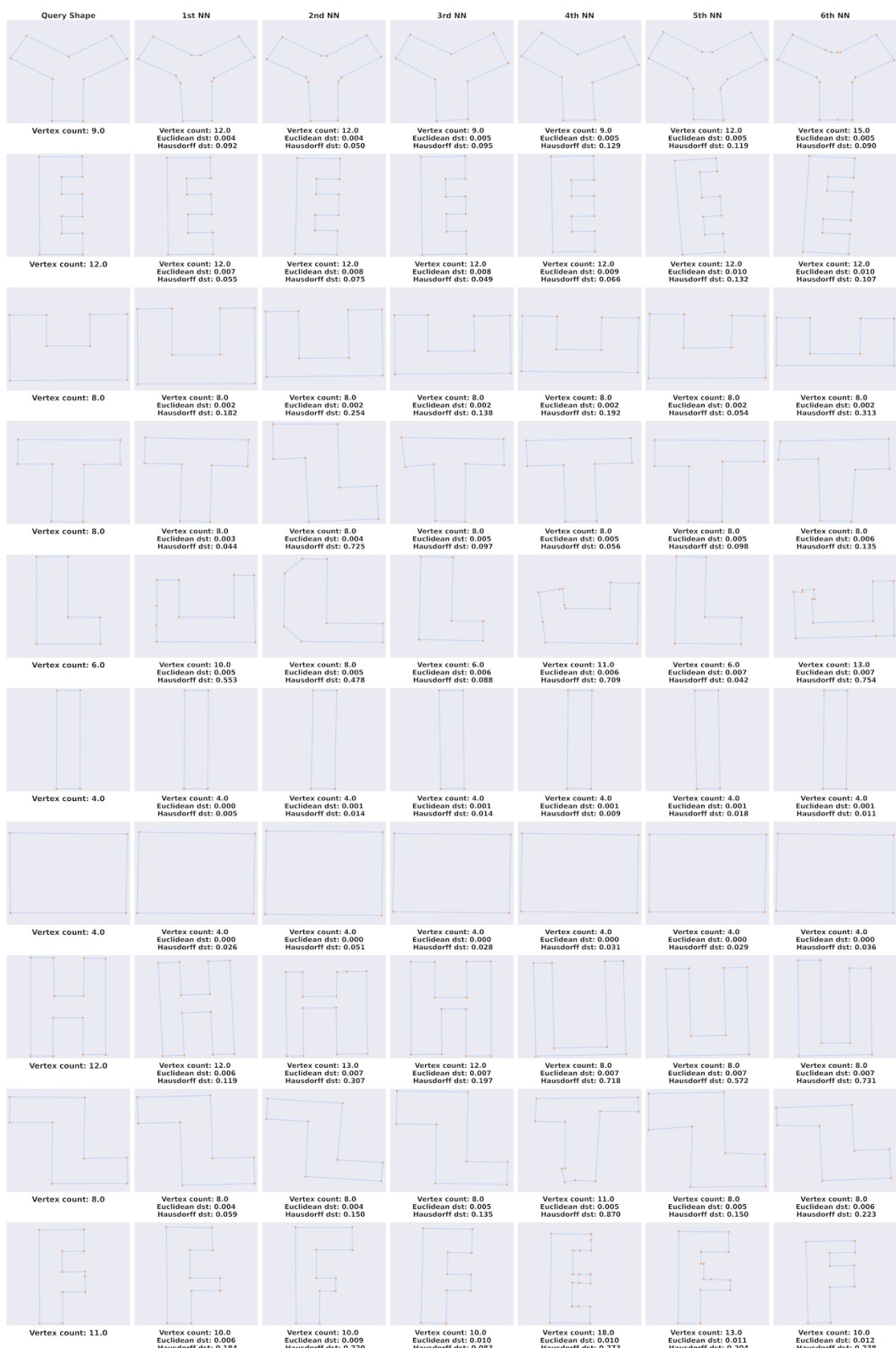

Figure 9: Polygon retrievals of NUFT on OSM-O using ten standard shapes as queries.

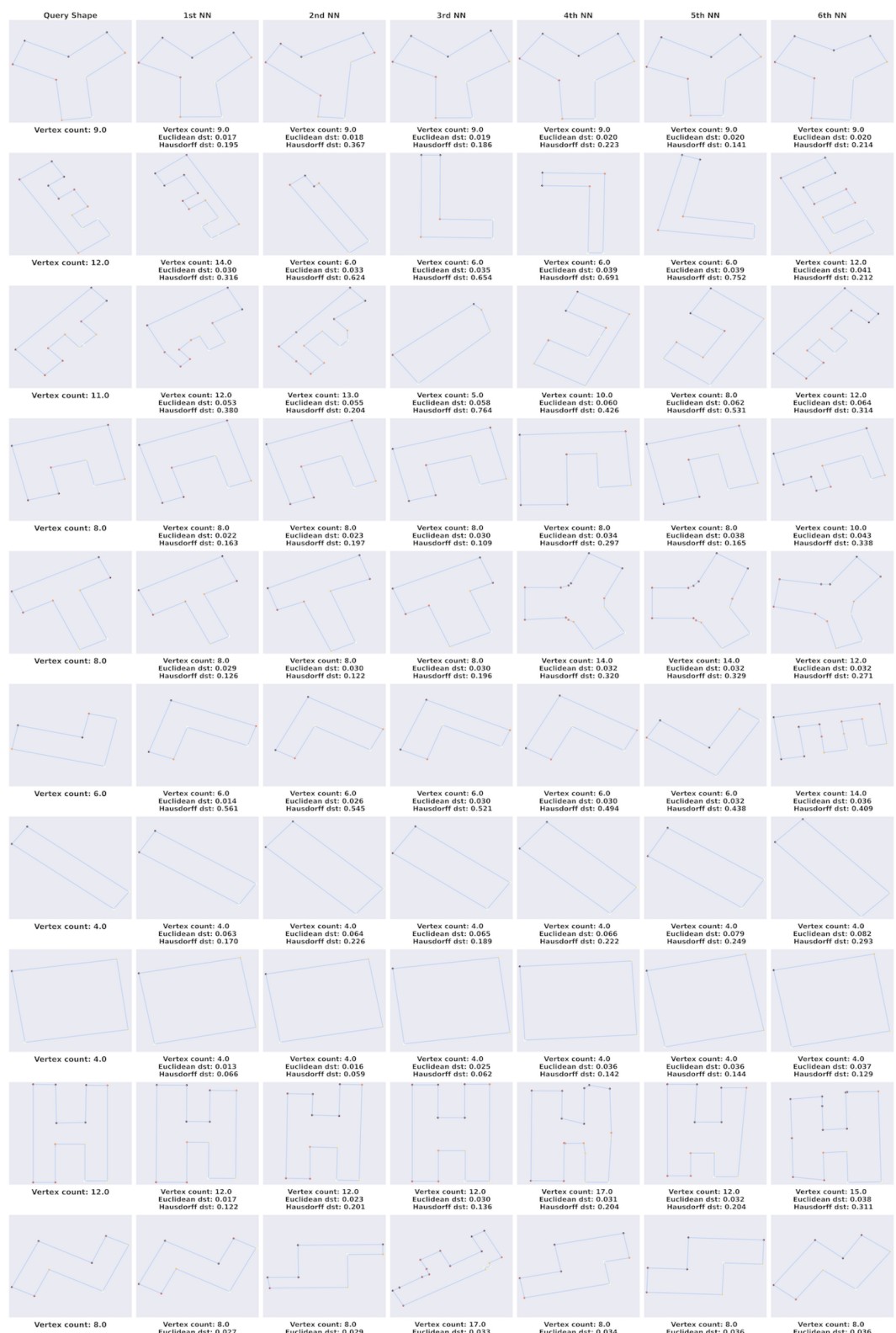

Figure 10: Polygon retrievals of GAE-GCN (baseline) on OSM-R using ten standard shapes as queries.

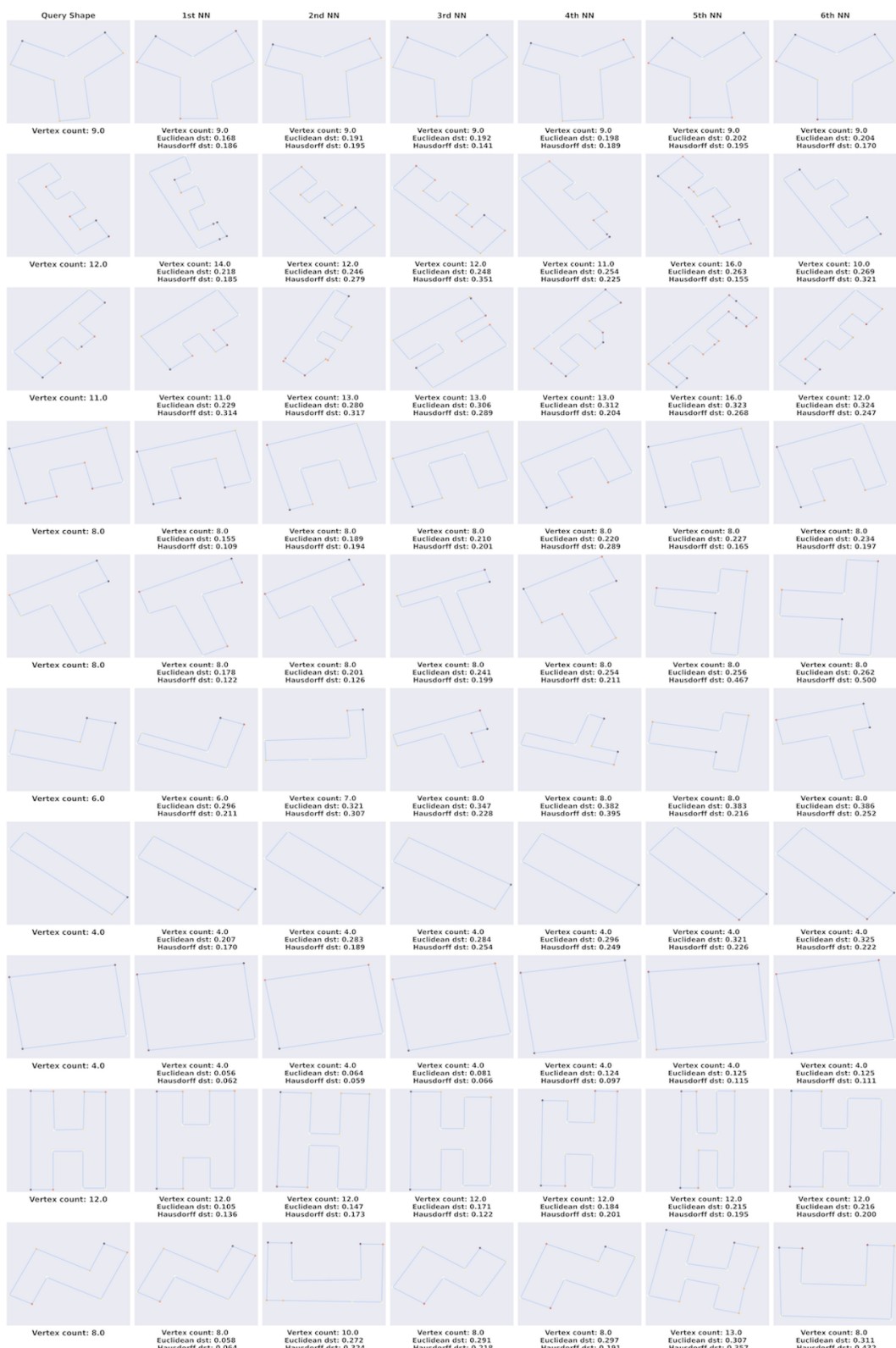

Figure 11: Polygon retrievals of CGAE-EdgeConv on OSM-R using ten standard shapes as queries.

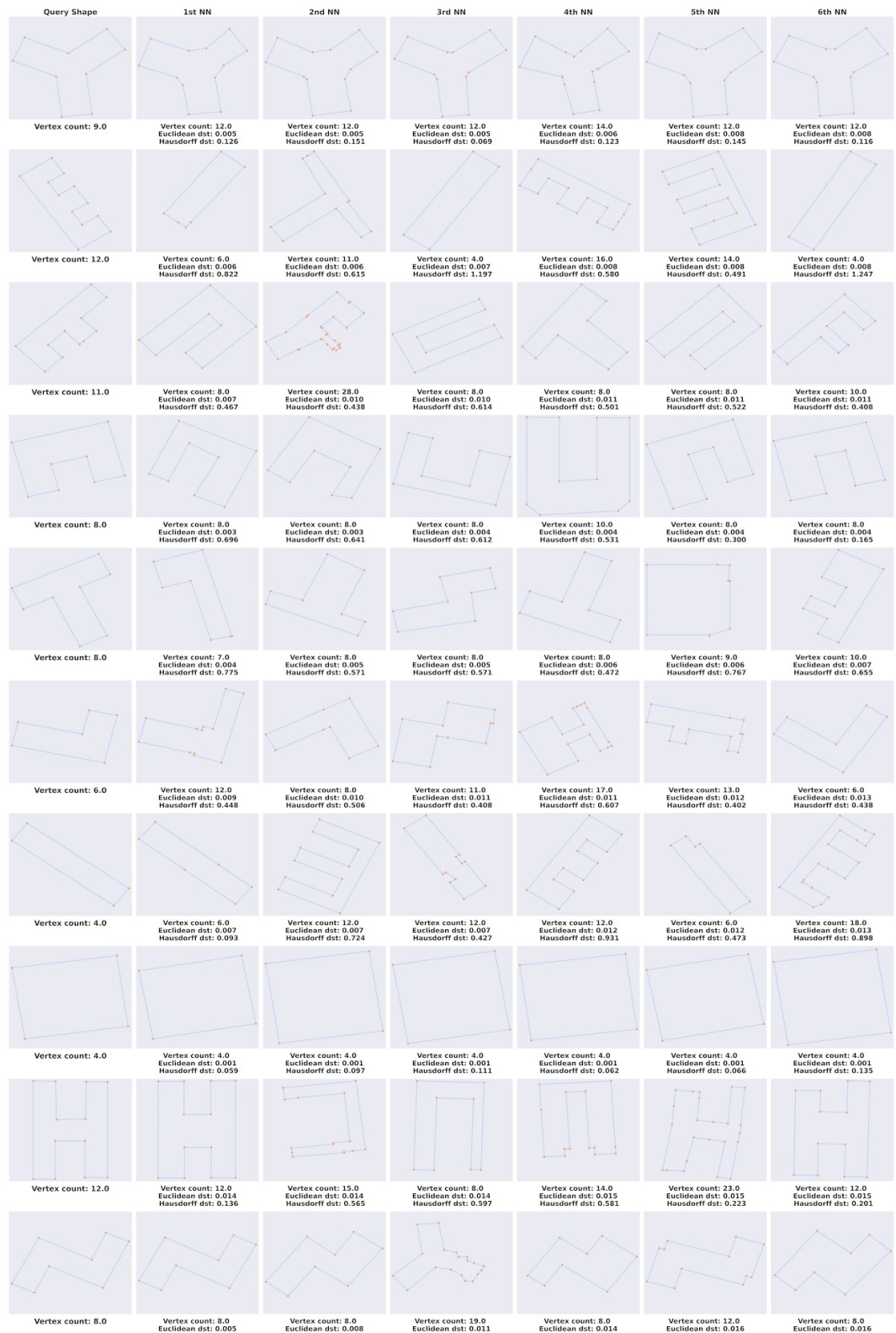

Figure 12: Polygon retrievals of NUFT on OSM-R using ten standard shapes as queries.

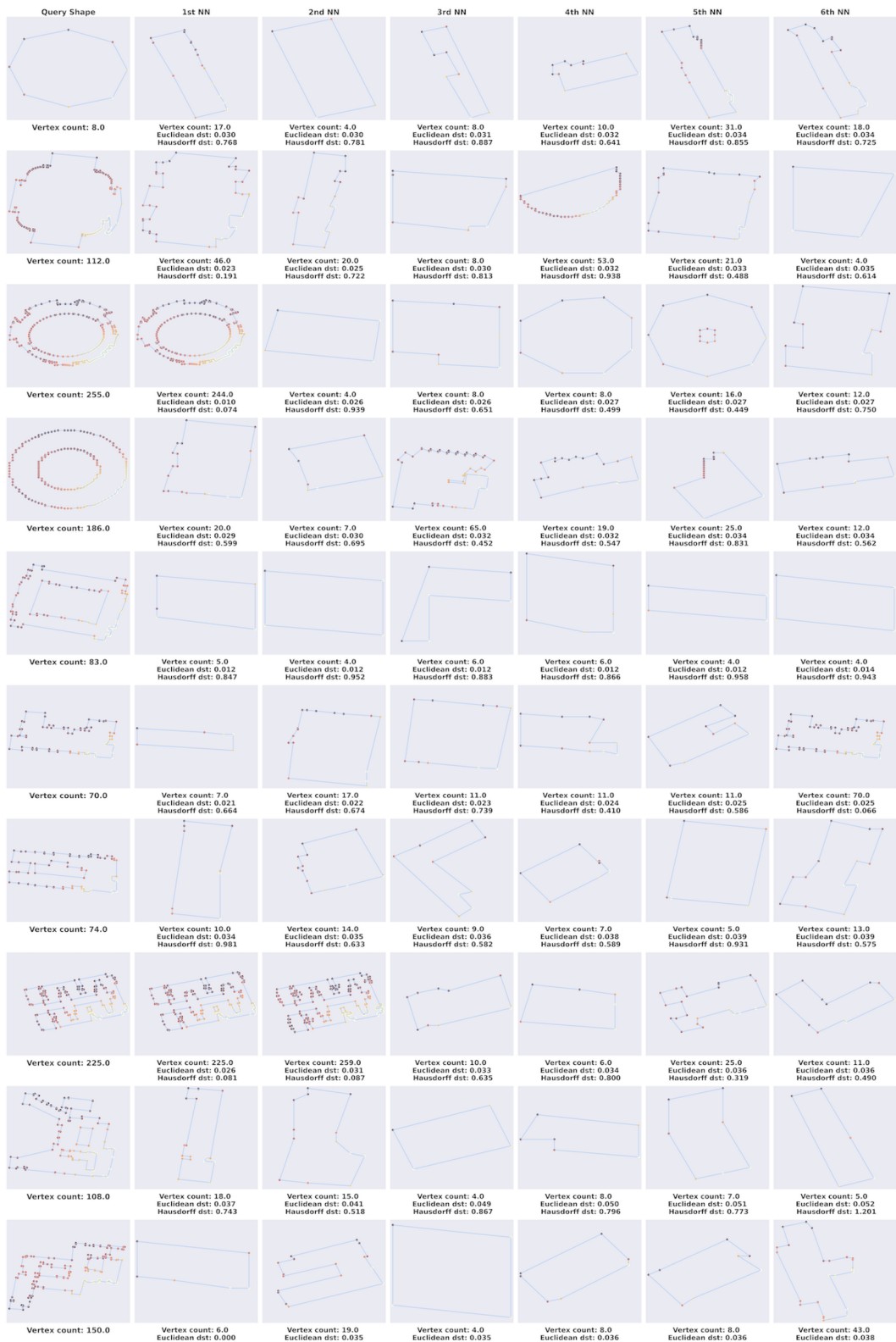

Figure 13: GAE-GCN (baseline) polygon retrieval on Melbourne building footprint dataset.

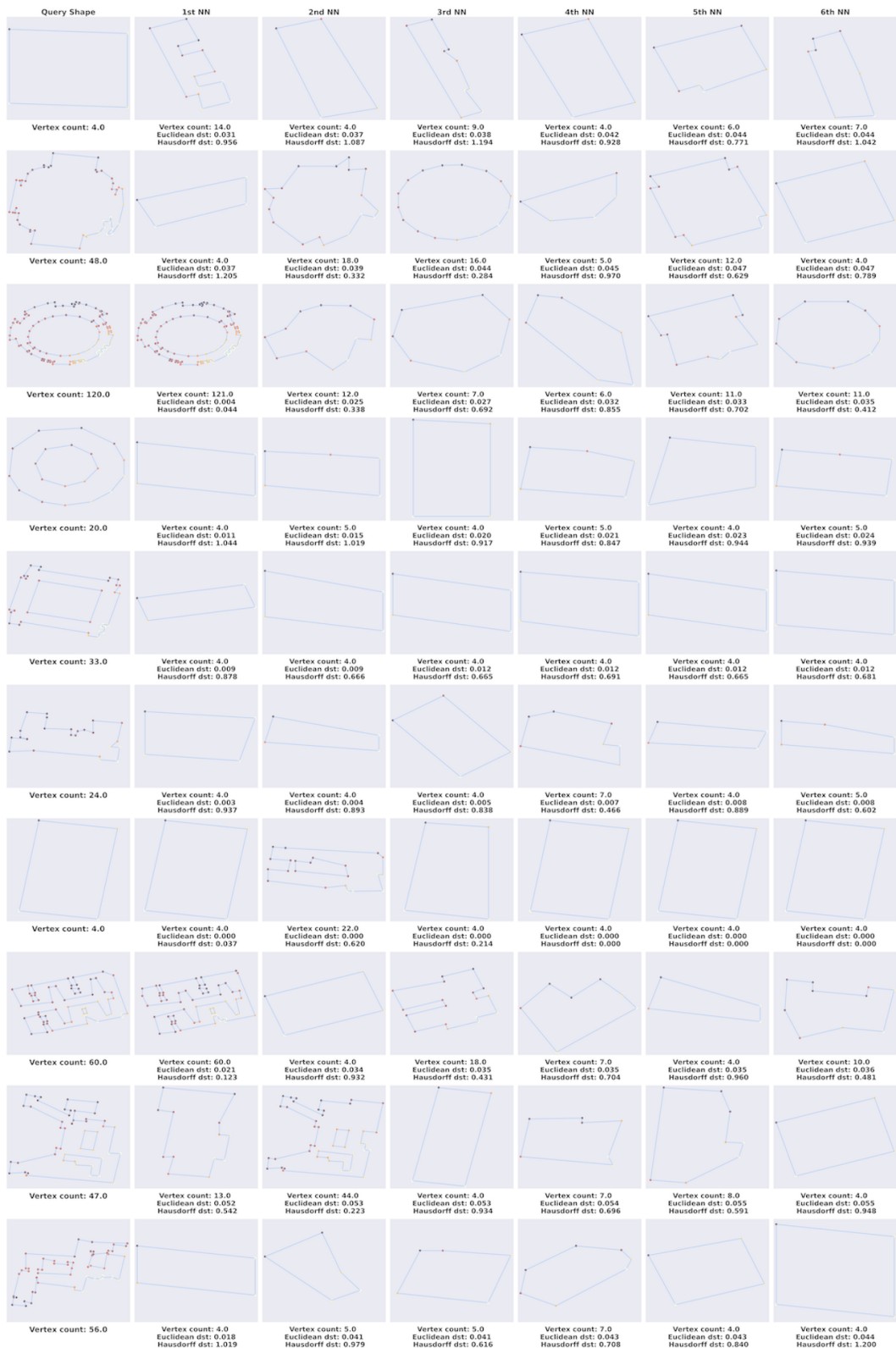

Figure 14: GAE-GCN (baseline) polygon retrieval on Melbourne building footprint dataset simplified by Douglas-Peucker algorithm (Douglas & Peucker, 1973) with $\epsilon = 0.00002$.

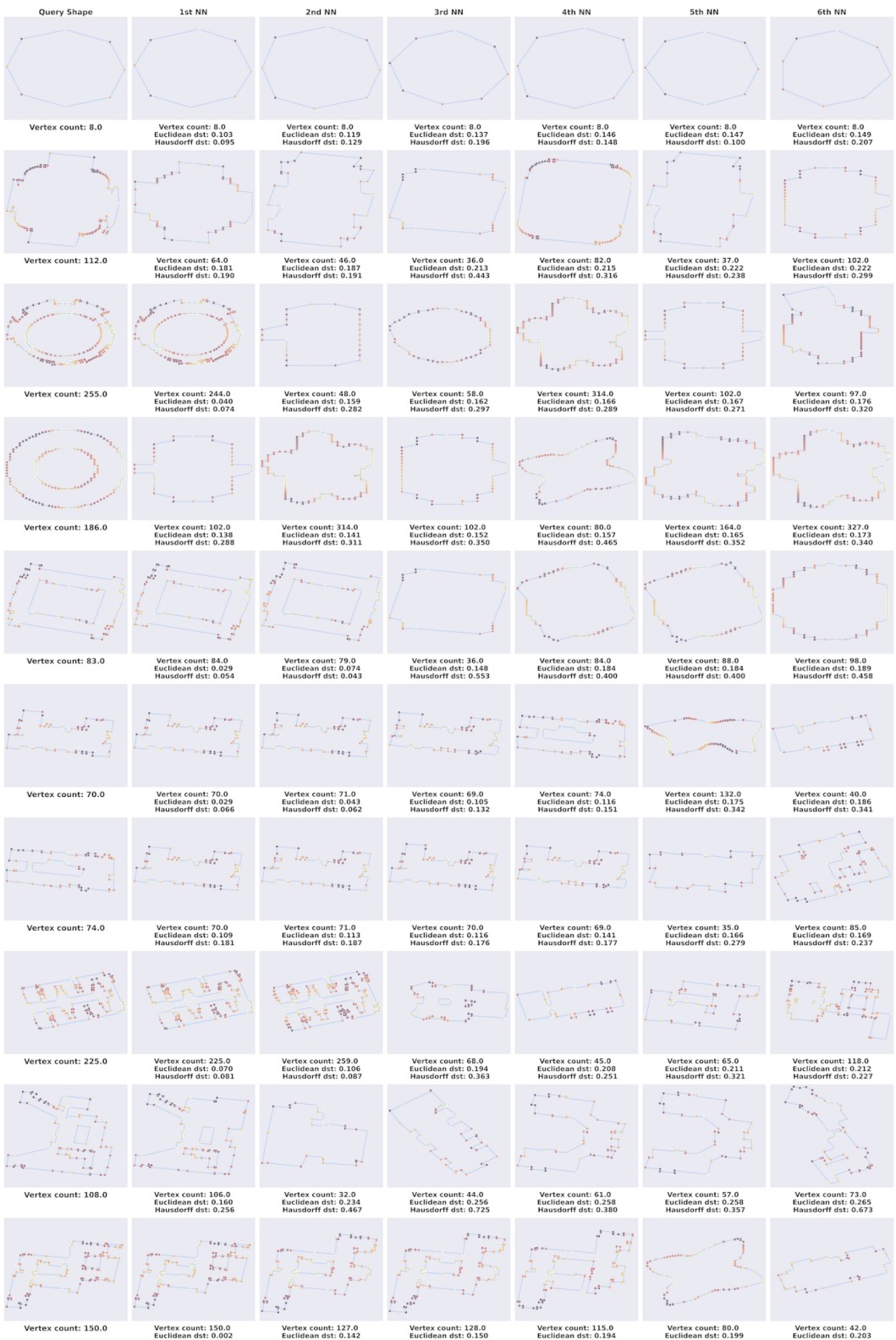

Figure 15: CGAE-EdgeConv polygon retrieval on Melbourne building footprint dataset.

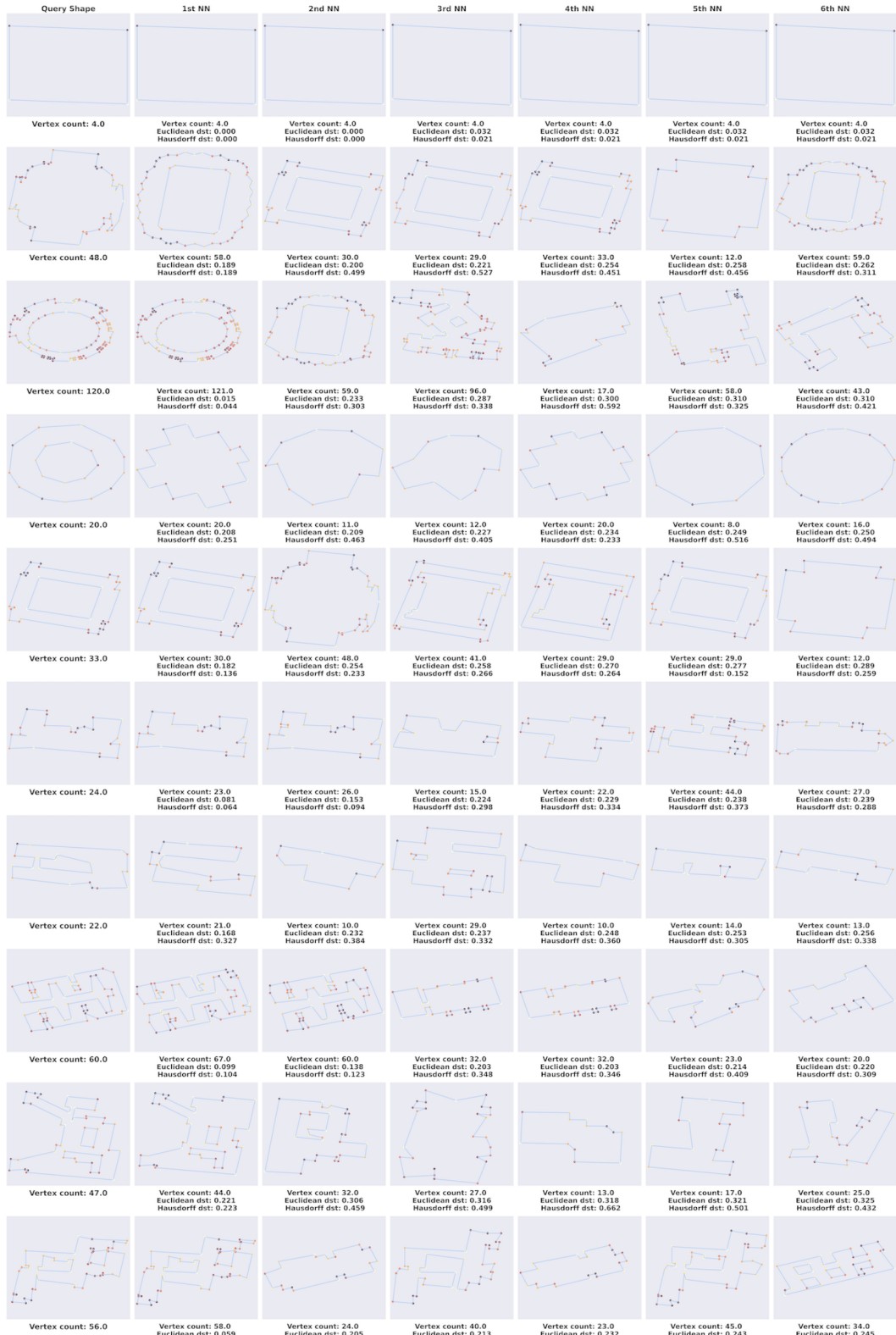

Figure 16: CGAE-EdgeConv polygon retrieval on Melbourne building footprint dataset simplified by Douglas-Peucker algorithm (Douglas & Peucker, 1973) with $\epsilon = 0.00002$.

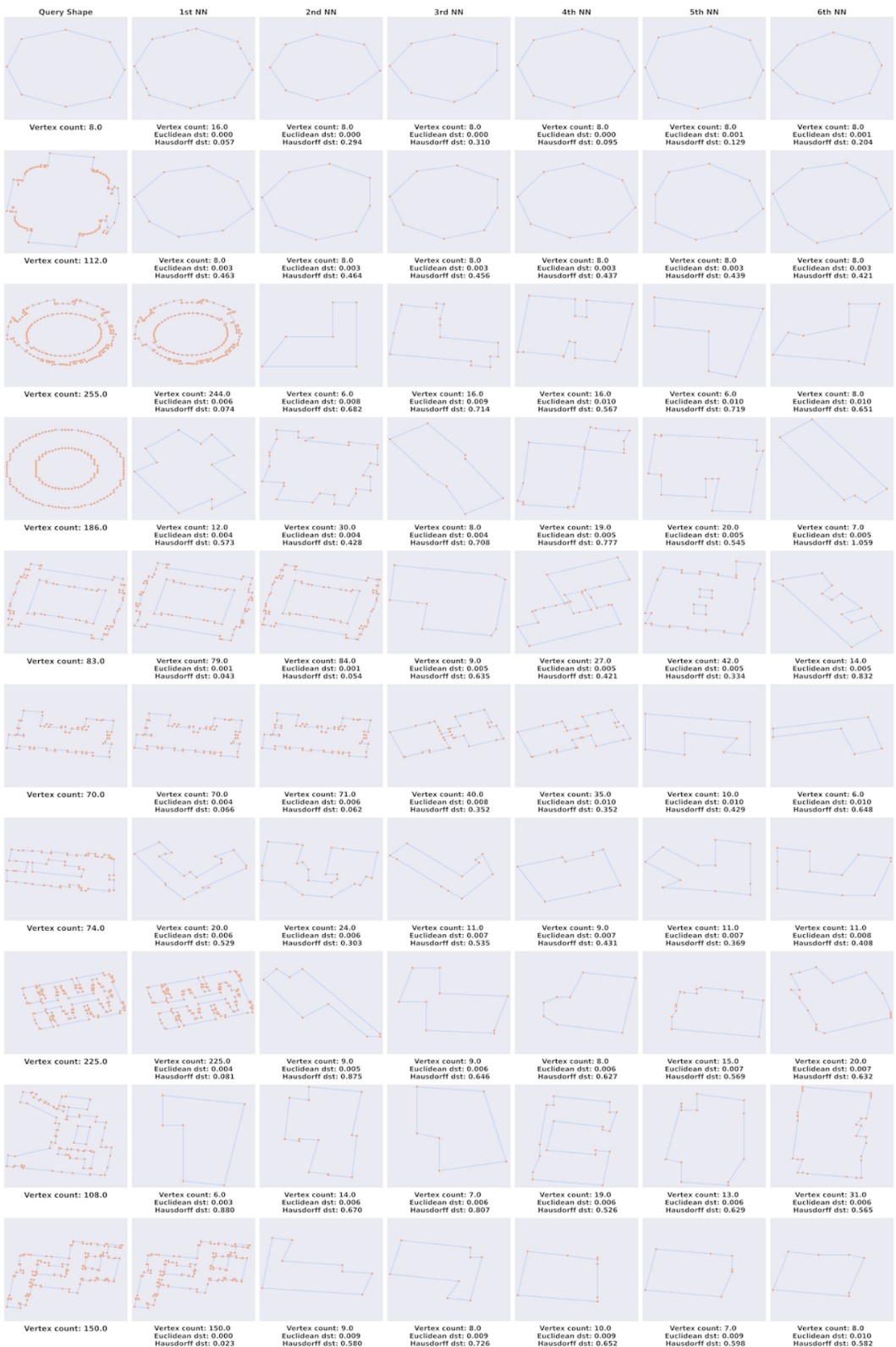

Figure 17: NUFT polygon retrieval on Melbourne building footprint dataset.

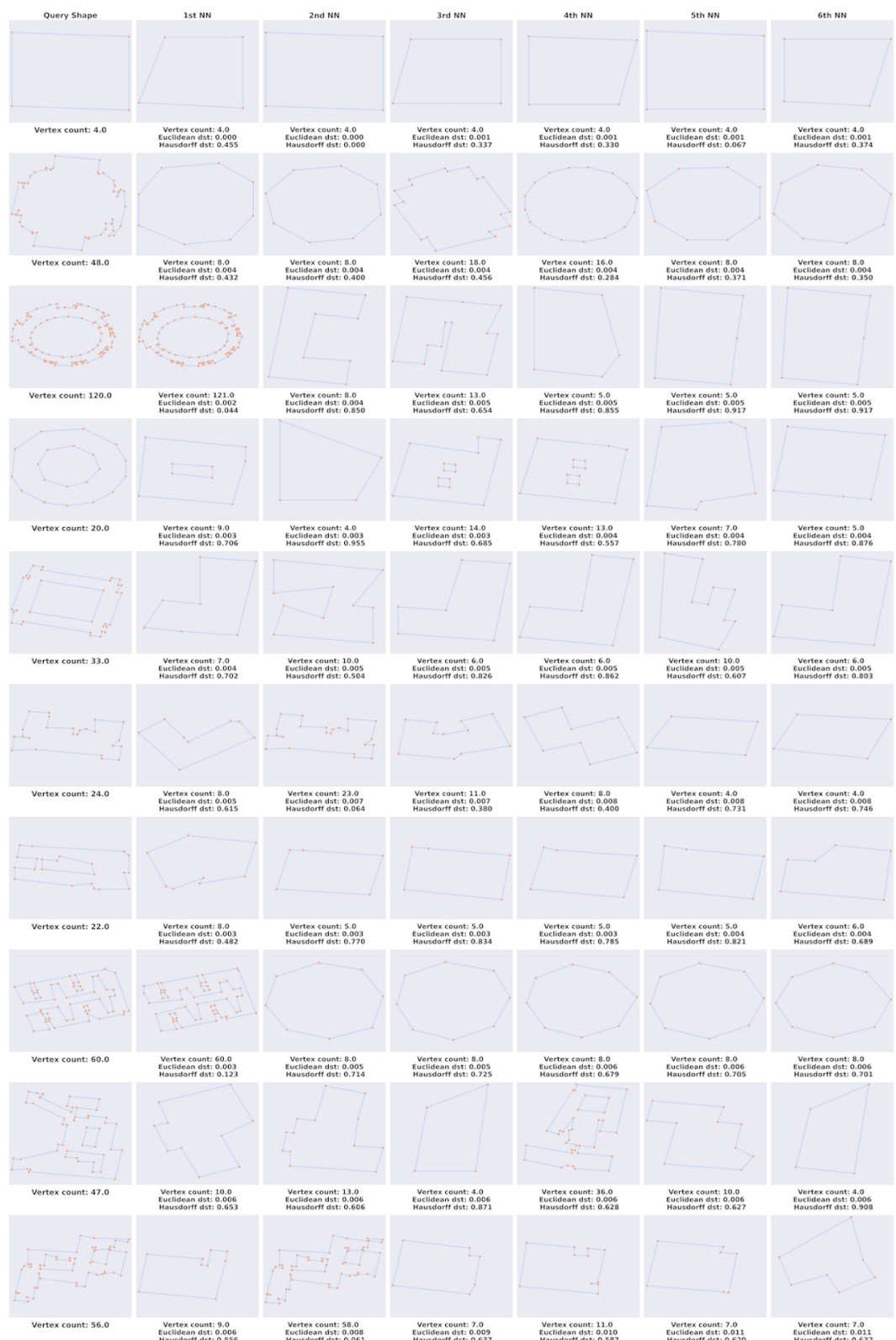

Figure 18: NUFT polygon retrieval on Melbourne building footprint dataset simplified by Douglas-Peucker algorithm (Douglas & Peucker, 1973) with $\epsilon = 0.00002$.

## F    NON-ML BASED BENCHMARKS

For completeness, we present the qualitative polygon retrieval performance on the Melbourne dataset, using the turning function (Arkin et al., 1991) in Fig. 19 (Melbourne) and 20 (Melbourne simplified); and using the Procrustes method (Goodall, 1991) in Fig. 21 (Melbourne) and 22 (Melbourne simplified).

### F.1    TURNING FUNCTION

The Turning function (Arkin et al., 1991) (TF) measures the tangent angles of polygons with a monotone function where function values increase for left-hand turns and decrease for right-hand turns on polygon boundaries. The accumulated function values (i.e., arc length) are used to compute a distance function (i.e., Euclidean distance) for comparing two simple polygons. This method is typically invariant under rotation, translation and scaling but sensitive to non-uniform noise.

Qualitative results in Fig. 19 (Melbourne) and 20 (Melbourne simplified) suggest that TF performs poorly on polygons without clear turning signals (Fig. 19, row 2 - 4) but performs relatively well on simplified polygons with salient turning signals (Fig. 20, row 3 and row 5).

### F.2    PROCRUSTES METHOD

Procrustes analysis (Goodall, 1991) discovers optimal geometric transformations (i.e., rotation, scaling and translation) that minimises the disparity (i.e., the sum of square of the point-wise differences) between two geometries. The Procrustes method retrieves geometries based on finding the nearest neighbours of queries with disparity as distance metric. The qualitative results of Procrustes are shown in Fig. 21 (Melbourne) and 22 (Melbourne simplified). Experimental results suggest that Procrustes is not suitable for handling complex polygons with holes in the retrieval task.

## G    PERSISTENT HOMOLOGY DIAGRAMS

To quantitatively measure the topological signatures of polygons (i.e., 0D connected components and 1D holes), we display the persistent homology diagram of the 10 query shapes of the Melbourne dataset in Fig. 23. Sub-figures (a) - (e) show persistent 1D ($H_1$) topological features, suggesting existence of holes in queries. Sub-figures (f) - (j) show persistent topological features in both $H_0$ and $H_1$, suggesting polygons with complex exterior boundaries and holes.

We refer the computation of persistent homology of data $\mathcal{X} \in \mathbb{R}^d$ to Zomorodian & Carlsson (2004).

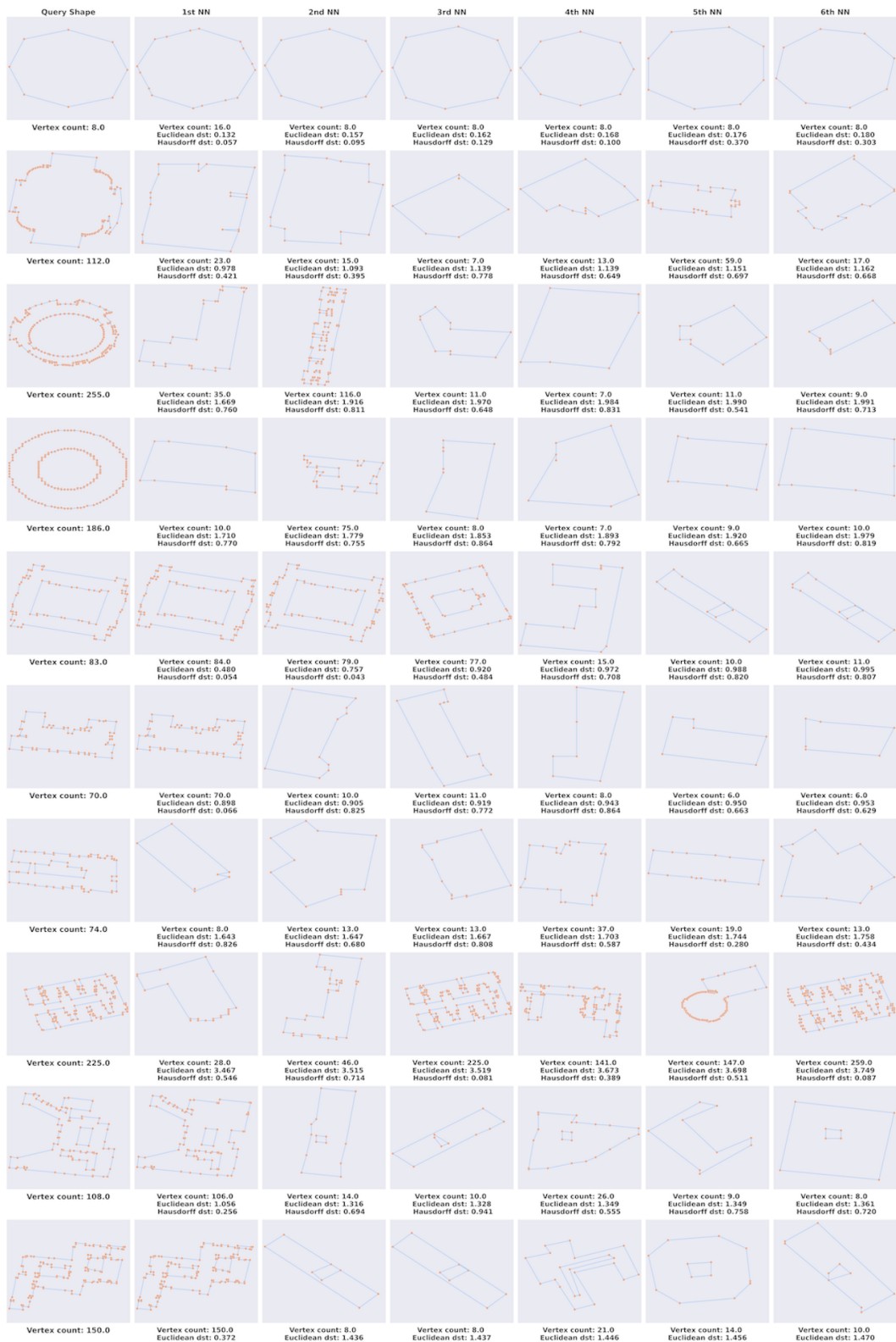

Figure 19: Polygon retrieval of Turning Function on Melbourne building footprint dataset.

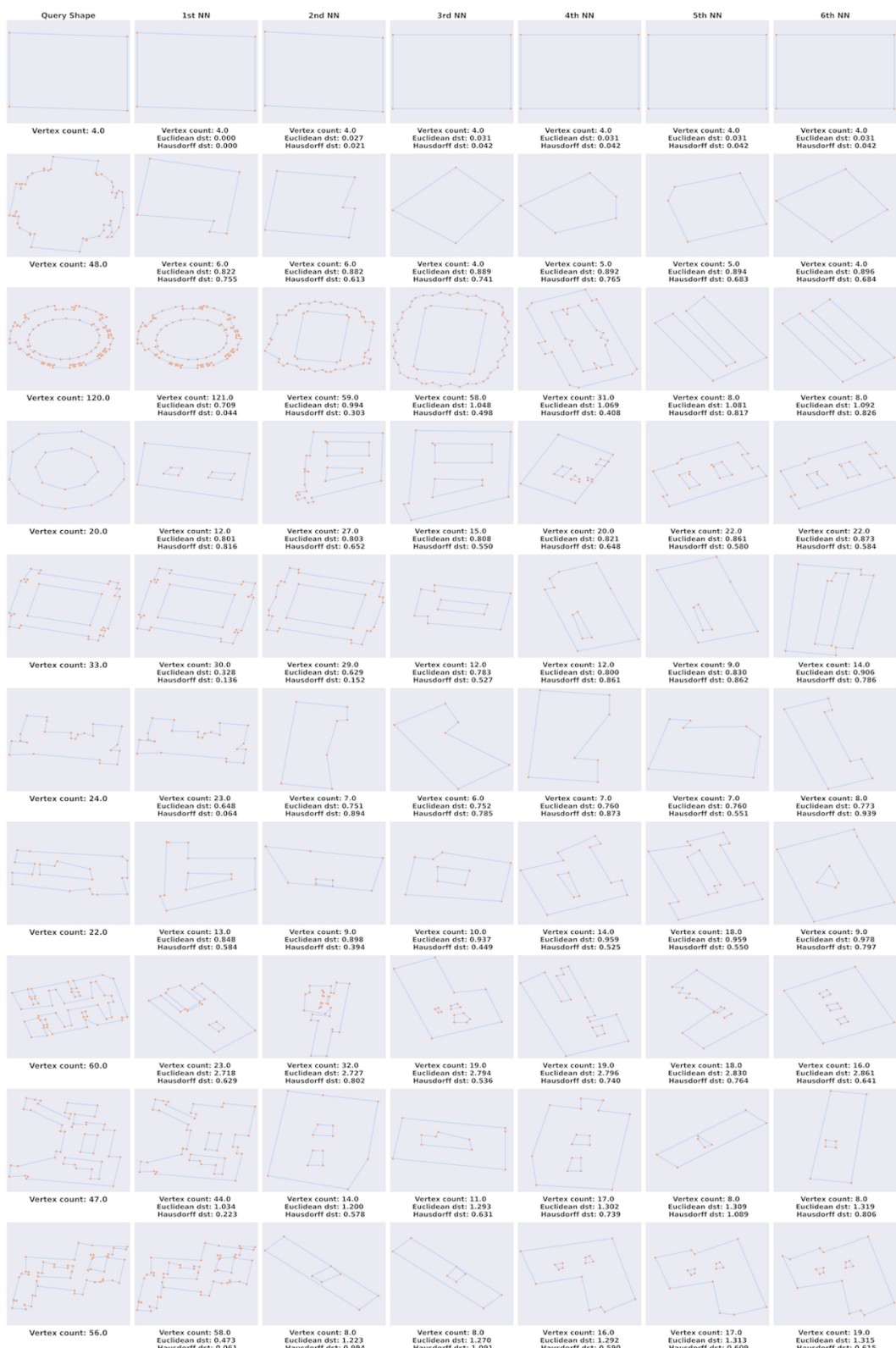

Figure 20: Polygon retrieval of Turning Function on Melbourne building footprint dataset simplified by Douglas-Peucker algorithm (Douglas & Peucker, 1973) with $\epsilon = 0.00002$.

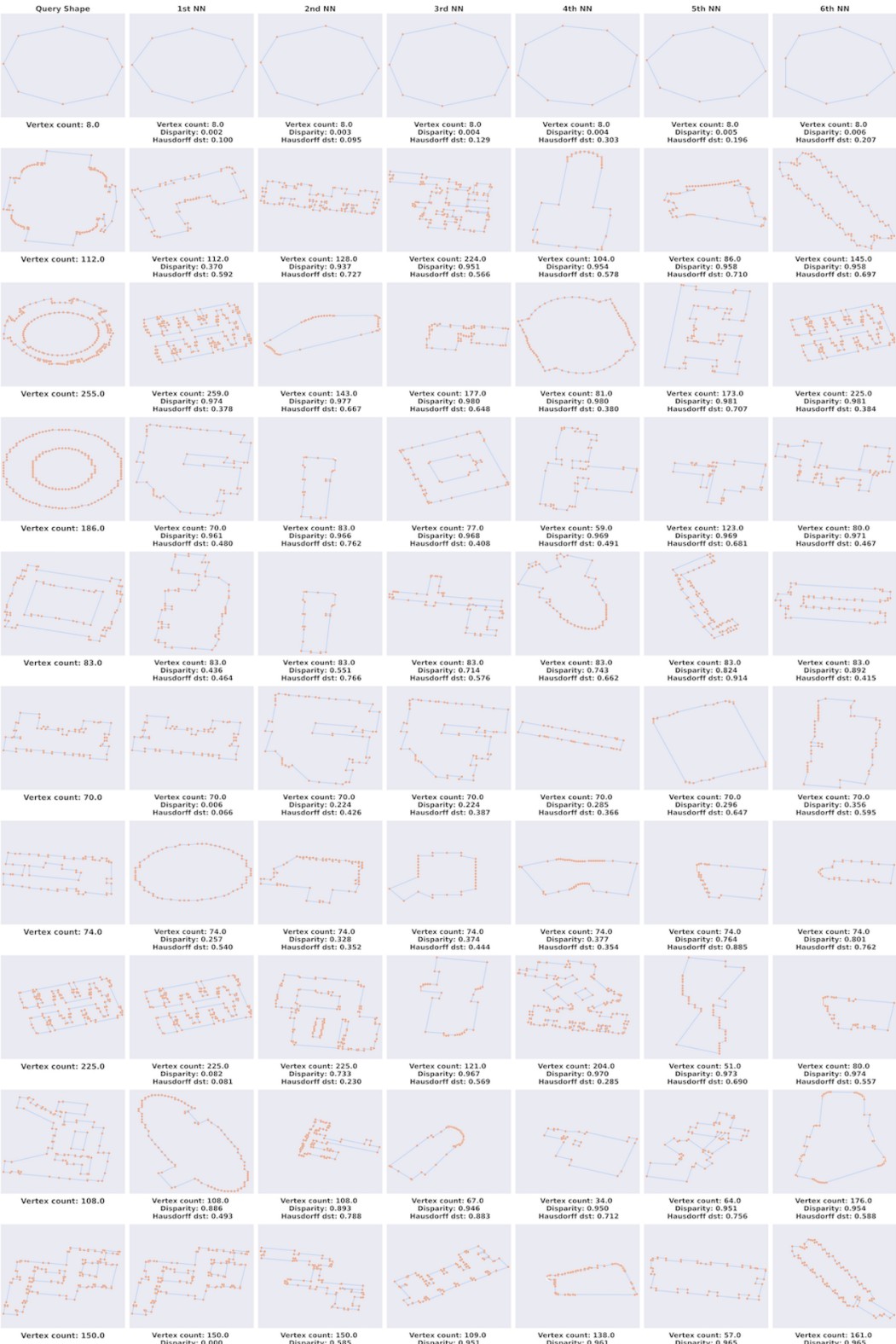

Figure 21: Polygon retrieval of Procrustes method on Melbourne building footprint dataset.

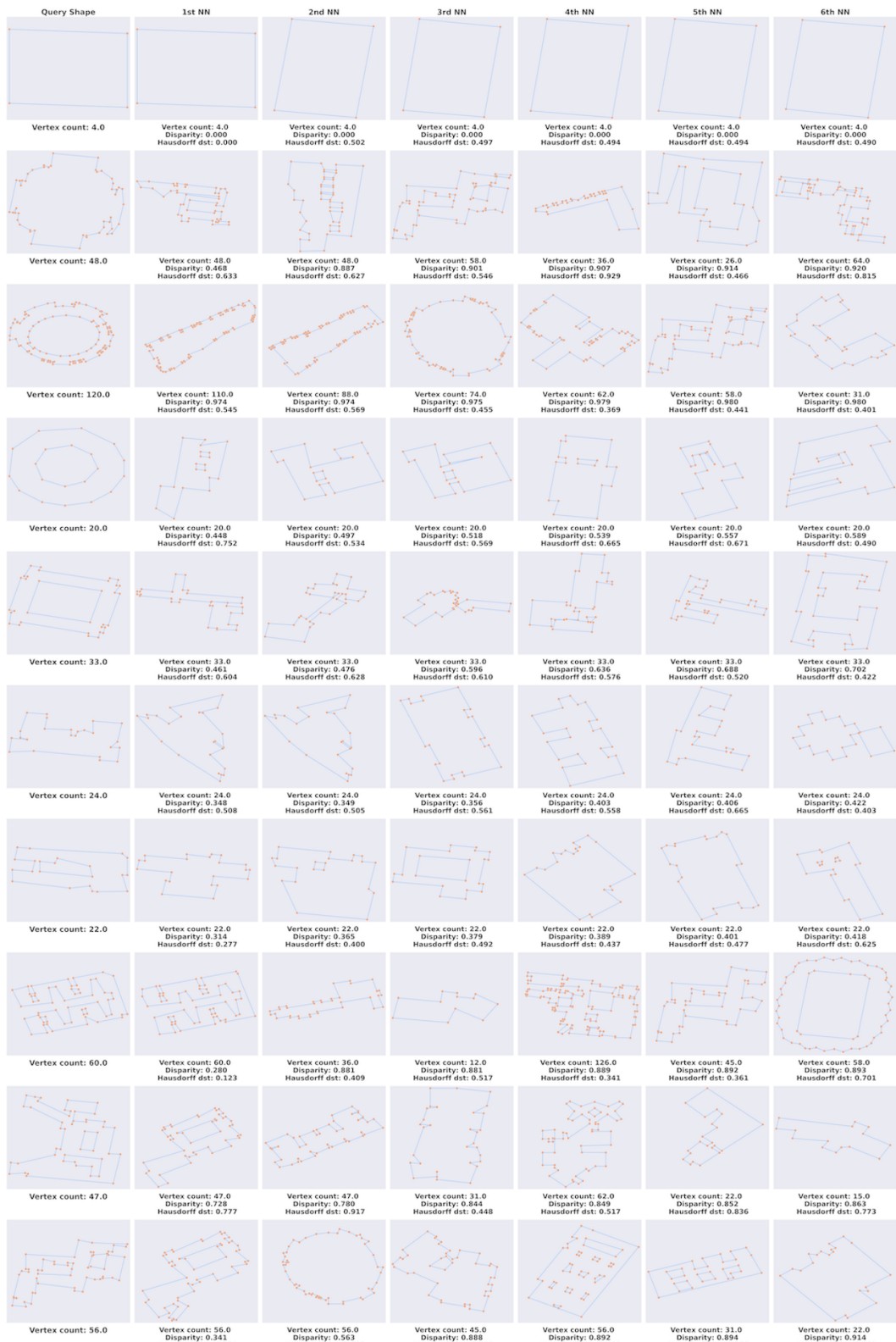

Figure 22: Polygon retrieval of Procrustes method on Melbourne building footprint dataset simplified by Douglas-Peucker algorithm (Douglas & Peucker, 1973) with $\epsilon = 0.00002$.

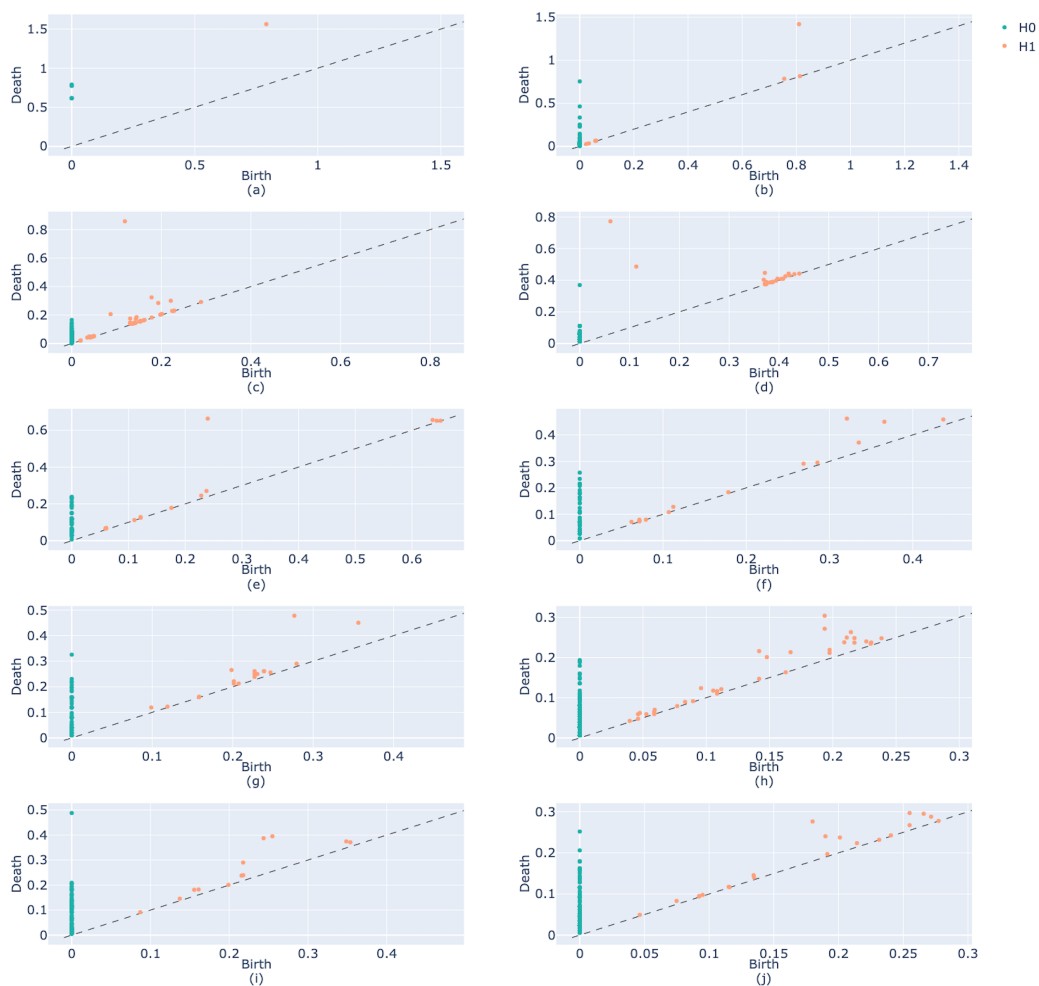

Figure 23: Persistent diagrams (a) - (j) of query shapes (row 1 -10) of Melbourne dataset. Persistent diagrams (Zomorodian & Carlsson, 2004) is a plotting of multi-set of points that summarise the topological signatures of irregular data $\mathcal{X} \in \mathbb{R}^d$ (i.e., 2D or 3D point sets). Birth (X-axis): time when data point $\mathcal{P} \in \mathcal{X}$ emerge. Death (Y-axis): time when data point $\mathcal{P} \in \mathcal{X}$ disappear. $H_0$: 0-th dimension topological feature that summarizes the number of connected component of data. $H_1$: 1-th dimension topological feature that summarizes the number of hole/loop in data.

