# OpenReview forum: "Contrastive Graph Autoencoder for Geometric Polygon Retrieval from Building Datasets"
_ICLR.cc/2024/Conference — Submitted to ICLR 2024_

### Official Review · Reviewer_sEzw · 2023-10-31

**Soundness:** 3 good
**Presentation:** 3 good
**Contribution:** 2 fair
**Rating:** 3
**Confidence:** 4

**Summary:**

This paper introduces a method to embed a polygon, which can be applied to polygon retrieval. The core of the method is to train an auto-encoder together with a contrastive learning loss. Since both reconstruction loss and contrastive learning loss doesn’t require human labels, their method can be unsupervised and scalable. The paper validates their method on multiple datasets including both synthetic and real-world datasets.

**Strengths:**

The proposed method doesn’t require any human labels and thus is scalable, particularly useful to the applications in the GIS domain where unlabelled data are sufficient due to crowd-source platforms like OpenOSM.

The proposed method also shows very promising results in the real world dataset.

**Weaknesses:**

The paper has several weaknesses which I’ll detail below:

The paper has a very limited comparison with baselines. The results only show the comparison between the different variants of the proposed methods, despite the fact that there are multiple existing works that tried to solve this problem like Yan et al as mentioned in the paper.

From my viewpoint, the impact of this paper on future research is quite limited. The proposed method is not new -- it's known that t contrastive learning together with autoencoder is better than one of them only, which limits its inspiration to future works. Also, the proposed method is not fully explored. For example, the most recent polygonal learning [1] lists two principled ways to encode the polygon geometry. The paper should also discuss, probably also try the better backbone and provide some insights.

Some important implementation details are missing, especially for the dataset preprocessing.
The reason why I think it’s important is because preprocessing might be able to help representation learning. For example, if the dataset is pre-aligned (rotation, scale, or translation), the representation learning would be much easier. Basically, different preprocessing might lead to very different performance. Thus, the paper should stress it in a very clear way.

Some claims of the paper are not fully validated. The paper claims that the proposed method is robust to the node count, rotations, etc. However, I’m not sure how the method achieves this. The paper doesn’t validate it using experiments or proof.

In short, I think the paper still requires a bit of work to polish before it is ready to be published. Although the presented method is effective in some datasets, the comparison is limited. Because of this, I’m not fully convinced that the proposed method is state-of-the-art. Also, the paper has other limitations that I detailed above.

But I’m glad to hear back from authors during rebuttal in case I misunderstand anything.

[1] Mai et al. Towards General-Purpose Representation Learning of Polygonal Geometries.

**Questions:**

Please address the questions and concerns that I have above.

---

> ### Author Response · Authors · 2023-11-15
> **Response to Reviewer sEzw**
>
> ## Reviewer sEzw
>
> **Comment sEzw-1** The paper has a very limited comparison with baselines. The results only show the comparison between the different variants of the proposed methods, despite the fact that there are multiple existing works that tried to solve this problem like Yan et al as mentioned in the paper.
>
>
> > **Response sEzw-1** We will provide additional results from the angular count (non-ML) method [Arkin, Esther M., et al.], used as one of  benchmarks in [Yan, Xiongfeng, et al.].
>
>
> **Comment sEzw-2** … the proposed method is not fully explored. For example, the most recent polygonal learning [1] lists two principled ways to encode the polygon geometry. The paper should also discuss, probably also try the better backbone and provide some insights.
>
> > **Response sEzw-2** We will attempt comparison with the methods of [Mai, Gengchen, et al.]. Note that in [Mai, Gengchen, et al.], the ResNet1D (a spatial domain polygon encoder, similar to [Yan, Xiongfeng, et al.]) and NUFT encoders (a fourier transform-based encoding method) are proposed for polygon embeddings to solve for shape classification of polygon geometries. The two encoding methods are tested on dataset MNIST-cplx70k, and we would like to point out that the Glyph dataset tested in our study likely presents a more challenging tasks to learning models (26 class vs. 10 class, as well as a less regular  (as not gridded) appearance of glyph outlines extracted directly from glyph representations and not raster images). We note that in the study of baseline [Yan, Xiongfeng, et al.], similar non-ML FT-based methods [Ai, Tinghua, et al.] are already tested on the same OSM dataset as here, but show lesser performance.
>
>
> **Comment sEzw-3** Some important implementation details are missing, especially for the dataset preprocessing. … For example, if the dataset is pre-aligned (rotation, scale, or translation), the representation learning would be much easier.
>
> > **Response sEzw-3** We note that the dataset preparation and testing of the present manuscript is fully documented in the reference provided as **Anonymous**. This manuscript is currently under double blind closed review, focusing on contributions distinct from this paper (comparison of architectures and data preprocessing in a classification task). We shall attempt to include as much detail here, enabling us to understand the dataset preparation applied here for one-shot retrieval, without jeopardizing the above review process.
> > To mitigate the concerns of **Reviewer sEzw** regarding the issue of data preprocessing, we update the anonymous Figshare URL provided in Reproducibility Statement section, adding the implementation code of data preparation of three datasets (Glyph, OSM and Melbourne). We also note that there is no pre-alignement between the training dataset and the arbitrary rotations of buildings in the real world.
>
>
> **Comment sEzw-4** Some claims of the paper are not fully validated. The paper claims that the proposed method is robust to the node count, rotations, etc. However, I’m not sure how the method achieves this.
>
> > **Response sEzw-4** Please, refer to **Response SxCs-2**.

---

> > ### Comment · Reviewer_sEzw · 2023-11-22
> > **Thanks for the feedback!**
> >
> > Thanks for the feedback and sorry for the late!
> >
> > However, sorry I'm not sure I'm fully convinced.
> >
> > First, as authors said, baselines are indeed missing, which I think might take quite a bit time to polish. Because the paper might ends up with a totally new story after these baselines are included.
> >
> > And then regarding proof about free of vertex count, I read the comments the authors provided to other reviewer. Honestly, I'm not very convinced. Because being free of vertex count is a bit strong claim whereas some example results provided during rebuttal are insufficient. To support this strong claim, I would suggest having more controlled experiments -- e.g., error analysis of model prediction for the different vertex count, or mathematically show the proof behind.
> >
> > Thus, I tend to keep my original rating.

---

> > > ### Author Response · Authors · 2023-11-22
> > > **Pre-final response to reviewer sEzw**
> > >
> > > Thank you for your feedback. We have added additional experiments demonstrating ML-based and non-ML baselines, as well as experiments on data with altered vertex counts ( generalisation to remove trivial vertices). We maintain that our model is **robust to vertex count change**, of course, within reason, hence we talk about *trivial vertices*, as is the practice in literature "(the overall nature of the shape must be preserved, and this is reflected in the Hausdorff distance). We have also refined our wording and toned down the statement of "free of vertex count" to *"robust to variable geometry vertex counts"* as we do in the remainder of the paper, apologies for the original statement in conclusions.
> > >
> > > We will be posting the updated version of the paper in the next few hours, and we sincerely hope that it will communicate our message stringently and to your satisfaction, enabling you to reconsider your scores.
> > >
> > > We really appreciate all the feedback that lifted the quality of our manuscript.

---

### Official Review · Reviewer_R5mS · 2023-10-31

**Soundness:** 3 good
**Presentation:** 3 good
**Contribution:** 3 good
**Rating:** 8
**Confidence:** 4

**Summary:**

The paper considers a geographic retrieval problem, querying a "map" with polygons to search for similar shapes. The paper considers the state of the art and prior work using graphical autoencoders (GAEs).  They seek to provide an improved solution that is robust to holes and translations of the polygons, problems that have affected prior solutions. They pick up a method that came from point cloud application, and incorporate contrastive learning to gain robustness. The writing is clear and the contribution and novelty are explained. The paper presents experiments and compares to the prior GAE solution as a baseline.

**Strengths:**

The work builds logically on the prior art and is well motivated to add robustness.  The contrastive learning framework allows the authors to introduce controlled deficits, adding or removing nodes and/or edges.

The authors work on known and available data sets and publish their code.

The comparison with the GAE as baseline seems fair.  The method is a logical extension and handles cases that the baseline does not.

**Weaknesses:**

The paper would benefit from an algorithm statement (say, in a Figure, or a flow diagram with equation references).  This should include a listing of all the learnable parameters, and their dimensions.  For example, after eqn (3)  and eqn (4) there are MLPs, but not clear where these are specified.

It isn’t clear if other (not necessarily ML-based) methods using geometry would work for these problems. The ideas of deformable templates and fitting are old, e.g., in the medical imaging literature.  However, the reviewer is not expert on these older methods.

The proposed method handles new cases, which is an important generalization, and shows better Hausdorf metrics for retrieving similar shapes in data.  However, it isn’t clear if this is now a “solved” problem or if more work is needed to generalize.

**Questions:**

Please say more about how eqn (6) is interpreted as a probability (or an estimate of a probability).

Did you consider different augmentation ratios r other than 20%?

From a practical perspective, are the results good enough to develop a tool that is broadly applicable?  What challenges remain?

Can you comment on the relation to this work: “Graph Contrastive Learning with Implicit Augmentations”, Arxiv, Nov 2022, and graph augmentation in general?

After eqn (9), should the losses all be weighted equally?  Should this be tuned for the application?

What are the tradeoffs with the k-NN, and choosing k?  How does this potentially help with a search, and does growing k imply more "incorrect" or bad cases to be reported?

Section 2.1: This seems intuitive but perhaps good to exactly define “linear ring”.

---

> ### Author Response · Authors · 2023-11-15
> **Response to Reviewer R5mS**
>
> ## Reviewer R5mS
>
> > **General Response** We will include an algorithm statement (possibly, as suggested, as a flow diagram with references to equations). And the listing of learnable parameters and their dimensions. This will specifically include the MLP specification for Eq 3 and 4, and a clarification for Eq 6.
>
>
> **Comment R5mS-1** Please say more about how eqn (6) is interpreted as a probability (or an estimate of a probability).
>
>
> > **Response R5mS-1** Eq. 6 calculates the reconstructed adjacency matrix $A$ from the node feature embedding $Z$. We apply a sigmoid activation ($\sigma$) to the square matrix of $Z$ to compute the probability between edges (i.e., ($z_i$, $z_j$) $\in$ $A$).
>
>
> **Comment R5mS-2** Did you consider different augmentation ratios r other than 20%?
>
>
> > **Response R5mS-2** We did not consider different augmentation ratios $r$ other than 20% in our study. To clarify, the augmentation ratio $r$ indicates, for a single polygon ($G$ = {$A$, $X$}), $r$% of node and edge are perturbed in graph augmentation (node -> node removal, edge -> edge addition). Rule of thumb is that after graph augmentation, the augmented graph $G$* should preserve its semantic information (qualitatively, geometric appearance). We deem $r$ = 20% a valid threshold for the study.
>
>
> **Comment R5mS-3** From a practical perspective, are the results good enough to develop a tool that is broadly applicable? What challenges remain?
>
>
> > **Response R5mS-3** We believe that these results are robust enough for a shape retrieval tool. We note that often individual building shapes are not deterministic of e.g., building function, and this may be revealed by analyzing the appearance of a broader embedding of buildings of similar shapes (typical British terrace houses, a sequence of elongated houses of a rectangular shape, adjacent and (topologically) touching each other) [Lüscher, P., Weibel, R. and Burghardt, D.]. Models interpreting shapes in their local embedding are subject of future research.
>
>
> **Comment R5mS-4** Can you comment on the relation to this work: “Graph Contrastive Learning with Implicit Augmentations”, Arxiv, Nov 2022, and graph augmentation in general?
>
>
> > **Response R5mS-4** We were not aware of this preprint, this has only appeared after our submission, thank you for bringing this to our attention. In *Anonymous*, we explore similar graph augmentation techniques, independently. We will cite and contrast our contribution.
>
>
> **Comment R5mS-5** After eqn (9), should the losses all be weighted equally? Should this be tuned for the application?
>
> > **Response R5mS-5** In our study, the GAE reconstruction and contrastive losses are weighted equally. $L = (\alpha) * (L_N + L_E) + (1 - \alpha) * L_C$, with $\alpha$ = 0.5. We have found that in the current experiments an equal weighting of GAE reconstruction loss ($L_N$ + $L_E$, Eq. 5 - 7) and contrastive loss ($L_C$, Eq. 8 - 9) was suitable but in future research we will consider tuning the weights.
>
> **Comment R5mS-6** What are the tradeoffs with the k-NN, and choosing k? How does this potentially help with a search, and does growing k imply more "incorrect" or bad cases to be reported?
>
>
> > **Response R5mS-6** We choose $k$ = 6  based on the convergence of the deficit of Haudorff distance between neighbors (i.e., Table 2, row 1, 1st NN -> 2nd NN (0.218 -> 0.238, 0.02 deficit) and 5th NN -> 6th NN (0.265 -> 0.271, 0.006 deficit)).
> In the t-SNE plot (Figure 2), we observe that both GAEs encode the latent embedding of polygons of similar geometric shapes (or of the same semantic class of alphabet letters) closer in latent space.
> Hence the shape extraction task is done by finding the k-nearest neighbors of embedding of query shapes. Intuitively, yes, growing $k$ implies retrieving more cases that are less similar as indicated in quantitative results (Table 1-3). Hausdorff distance of 1st NN is generally lower than 6th NN.
>
>
> **Comment R5mS-7** Section 2.1: This seems intuitive but perhaps good to exactly define “linear ring”.
>
>
> > **Response R5mS-7** According to Open Geospatial Consortium. Opengis simple features specification for SQL revision 1.0, 2003, the linear ring is defined as “simple geometries [aka, without self intersections], where rings are defined by a series of points with linear interpolation between points, and the first and last point are identical.” We will add this to the manuscript.

---

### Official Review · Reviewer_SxCs · 2023-11-01

**Soundness:** 2 fair
**Presentation:** 3 good
**Contribution:** 2 fair
**Rating:** 5
**Confidence:** 3

**Summary:**

The study presents a novel approach, the Contrastive Graph Autoencoder (CGAE), to effectively retrieve polygons with similar shapes from geographic maps, a task that has been challenging due to the complexity of polygon geometries and their susceptibility to transformations like rotation and reflection.

CGAE, utilizing advanced graph message-passing, feature augmentation, and contrastive learning, excels in encoding distinctive latent representations of polygon shapes, facilitating more accurate geometry retrieval. CGAE outperforms traditional graph-based autoencoders (GAEs) through experiments with real-world building map datasets.

**Strengths:**

1. In contrast to traditional models and state-of-art learning-based graph autoencoders, CGAE is independent of polygonal vertex counts
2. CGAE is capable of retrieving polygons with or without holes
3. CGAE is robust to polygon reflections and rotations
4. CGAE can effectively generalize to large polygon datasets

**Weaknesses:**

Lack of important experimental results that support the conclusions. See questions for details.

**Questions:**

1. The author claimed the proposed method could retrieve polygons with holes better than existing methods, but did not provide quantitative metrics like persistent diagram whose 0-th dimension topological feature represents the number of component and 1-th dimension topological feature represents number of holes.

2. The author claimed the proposed method is free of polygonal vertex counts which is a major benefit over the traditional methods and other graph autoencoders, but I cannot find any experiment results in the paper that can support such claim. Basically CGAE is close to GAE in terms of architecture so why CGAE can scale significantly better than GAE? I think the author should provide more evidences and arguments to this point since it is a critical contribution claimed by the author.

3. Why is CGAE robust to polygon rotations and reflections given that the proposed contrastive loss mainly focuses on local perturbation?

4. Can you clarify this sentence

` CGAE generalizes effectively the geometric information learned from simple polygons to complex shapes, demonstrating a desirable model property, i.e., decoupling of shape detail (i.e., polygon vertex count) from classification accuracy.`

I'm a bit confused about what classification means here.

---

> ### Author Response · Authors · 2023-11-15
> **Response to Reviewer SxCs Q 1 - 2**
>
> **Comment SxCs-1** The author claimed the proposed method could retrieve polygons with holes better than existing methods, but did not provide quantitative metrics like persistent diagram whose 0-th dimension topological feature represents the number of component and 1-th dimension topological feature represents number of holes.
>
>
> > **Response SxCs-1** Thanks for the suggestion. We update the manuscript by generating persistent diagrams for ten query geometries from the Melbourne dataset (as demonstrated in Appendix, Figure 10-11), showing the number of components (indicated by 0-th dimension topological features) and number of holes (indicated by 0-th dimension topological features). Note that the polygon geometries in Melbourne dataset (in total of 37139 samples) contains varying number of holes and are difficult to quantify as compared to other synthetic datasets (such as Glyph dataset and MNIST-cplx70k dataset in [Mai, Gengchen, et al.]).
>
> **Comment SxCs-2** The author claimed the proposed method is free of polygonal vertex counts which is a major benefit over the traditional methods and other graph autoencoders, but I cannot find any experiment results in the paper that can support such claim. Basically CGAE is close to GAE in terms of architecture so why CGAE can scale significantly better than GAE? I think the author should provide more evidences and arguments to this point since it is a critical contribution claimed by the author.
>
> > **Response SxCs-2** Recall that our models are trained on the synthetic Glyph dataset and tested on real-world building footprints (OSM and Melbourne dataset). The number of vertex counts of the real-world building footprints vary compared to the synthetic dataset.
> The experimental results demonstrate how the current method is free of vertex count dependency are shown in the Appendix, Figures 10 - 11. Take as example query shapes 3 and 4 ( complex circular polygons with a single hole). The baseline GAE extracts polygons with over-simplified exteriors, which are not geometrically similar to the queries. CGAE, oppositely, can extract polygons with complex exteriors that are similar to the queries. In queries 5 - 7, GAE and CGAE fetch polygons for complex rectangular queries with or without holes. CGAE is capable of extracting rectangular counterparts for complex rectangular queries whereas GAE only identifies over-simplified rectangular polygons. We finally investigate CGAE’s capability of retrieving polygons for complex queries with multiple holes.
> We update the manuscript by adding additional experiment results in Appendix, Figures 12 - 13. The two figures demonstrate the qualitative results of proposed CGAE and baseline GAE on simplified polygon geometries (with trivial vertices removed) by the topology-preserving Douglas-Peucker algorithm [Saalfeld, Alan.].
> We also address the comment  “CGAE scales better than GAE”. We do *not* make any such claim - they are equivalent in how they scale, but CGAE is more robust and accurate.
> Comparing Figures 6-7 (original sample from OSM dataset in [Yan, Xiongfeng, et al.]) to Figures 8-9 (rotated sample from OSM dataset in [Yan, Xiongfeng, et al.]), we observe how the baseline GAE is sensitive to rotations/reflections of buildings, failing to extract similar shapes when query shapes are rotated. The empirical quantitative results (in Table 1 Glyph-R, and Table 2 OSM-R) further support this claim.

---

> ### Author Response · Authors · 2023-11-15
> **Response to Reviewer SxCs Q 3 - 4**
>
> **Comment SxCs-3** Why is CGAE robust to polygon rotations and reflections given that the proposed contrastive loss mainly focuses on local perturbation?
>
>
> > **Response SxCs-3** We attribute the CGAE’s robustness to geometric transformations (i.e., rotations and reflections) to two aspects: 1. The Message-passing layers. Empirical results in the ablation study (Table 1, Glyph-R and Table 2, OSM-R) support that argument. GIN and EdgeConv (message-passing backbones) improves model performance on both datasets; and 2. Graph reconstruction based on both node-wise and edge-wise features (Eq. 5-7). In particular, compared to baseline GAE, the proposed CGAE reconstructs graph features from the perturbed node-wise and edge-wise features given by graph augmentations, forcing the graph autoencoder to reconstruct proper graph embedding from the perturbed latent embedding. This is supported by the quantitative results of ($L_N$ + $L_E$) in ablation study (Table 1-2) .
>
>
> **Comment SxCs-4** Can you clarify this sentence “CGAE generalizes effectively the geometric information learned from simple polygons to complex shapes, demonstrating a desirable model property, i.e., decoupling of shape detail (i.e., polygon vertex count) from classification accuracy.” I'm a bit confused about what classification means here.
>
>
> > **Response SxCs-4** We rephrase the sentence into “CGAE generalizes effectively the geometric information learned from simple polygons to complex shapes, demonstrating a desirable model property, i.e., decoupling of shape detail (i.e., polygon trivial vertex count.”) from shape retrieval accuracy.

---

### Official Review · Reviewer_sWPL · 2023-11-02

**Soundness:** 4 excellent
**Presentation:** 3 good
**Contribution:** 2 fair
**Rating:** 5
**Confidence:** 4

**Summary:**

This paper proposes a way to encode building footprint polygons using a message-passing graph encoder trained with node, edge and contrastive losses. The method improves upon the cited prior work (Yan et al.) by handling polygons with holes, being more discriminative because of the additional losses, and being more robust to noise via additional augmentations.

**Strengths:**

The method is well-argued and technically sound, the technical contributions over the cited prior work (Yan et al.) are reasonable, and the results are good.

**Weaknesses:**

I am a little unsure about the magnitude of the contribution here. By and large, it rests on three features:

1. A new (compared to Yan et al., but not new overall) GNN backbone that can accommodate graphs with multiple connected components
2. An edge-preservation loss.
3. A contrastive loss.

These are perfectly reasonable and I have no specific criticisms of these choices. But I am not sure there is any insight that is specific to _polygons_, and as a result the method ends up as a way to retrieve arbitrary graphs (I think) and hence ends up in a much larger solution space of prior work on retrieval from collections of graphs, e.g.

Li et al., "Graph Matching Networks for Learning the Similarity of Graph Structured Objects", ICML 2019

or (for 3D layouts) Li et al., "GRASS: Generative Recursive Autoencoders for Shape Structures", SIGGRAPH 2017

Minor:
- Several parts of the text have "massage-passing" instead of "message-passing"

**Questions:**

Apropos the comments above: what in the method is specific to polygons? Would this work for arbitrary graphs? Could improvements be made by considering the graphs are specifically non-intersecting polygon boundary loops?

The authors evaluate on the Glyph benchmark but only compare to ablated versions corresponding to a baseline GAE method obtained by ablation (with different backbones). There must be other relevant baseline methods for graph/polygon/glyph/sketch retrieval?

Is there a specific need, in the studied domain of building footprints, to consider the precise topology of the input polygons? E.g. if a straight boundary segment is subdivided into several smaller segments by inserting vertices, the footprint is geometrically the same. But the descriptor will presumably change. Is this desired behavior or not? I understand the authors do additional augmentation to be robust to geometric and topological noise, but this sort of variation appears to be beyond the scope of that augmentation.

In this context, why not just encode the raster footprints instead of the non-regular vector representations? Building footprints are surely simple enough that the advantages of vector representations for complex geometry don't really apply.

---

> ### Author Response · Authors · 2023-11-15
> **Response to Reviewer sWPL**
>
> ## Reviewer sWPL
>
> **Comment sWPL-1** What in the method is specific to polygons? Would this work for arbitrary graphs? Could improvements be made by considering the graphs are specifically non-intersecting polygon boundary loops?
>
> > **Response sWPL-1** In our submission we focus on the retrieval of *simple* (as in, without self loops , intersections, and tangents) polygons *embedded* in 2D space, with linear segment edges. This provides a strong inferential bias to the retrieval method. The generalisation to arbitrary graphs ( e.g., social networks, etc) that only capture topology, but not geometry is not assured, and not the focus here. In such arbitrary graphs the planarity is not guaranteed. Here, the spatial embedding of the vertices is as important as the ring topology of the vertex connectivity. Hence we assume that any graphs where this method contributes should include, broadly speaking, geometry (shapes) as well as topology (the location of the vertices provides an inferential bias).
>
> **Comment sWPL-2** The authors evaluate on the Glyph benchmark but only compare to ablated versions corresponding to a baseline GAE method obtained by ablation (with different backbones). There must be other relevant baseline methods for graph/polygon/glyph/sketch retrieval?
>
> > **Response sWPL-2** Please, see also Response **sEzw-1**.
>
> **Comment sWPL-3** Is there a specific need, in the studied domain of building footprints, to consider the precise topology of the input polygons? E.g. if a straight boundary segment is subdivided into several smaller segments by inserting vertices, the footprint is geometrically the same. But the descriptor will presumably change. Is this desired behavior or not?
>
> > **Response sWPL-3** We design our system to mimic the perspective of human perception of shapes. This is what is here captured by “trivial vertex count independence” (i.e., colinear vertices taht do not add “information). See response **SxCs-2** and **sEzw-4**.
>
>
> **Comment sWPL-4** In this context, why not just encode the raster footprints instead of the non-regular vector representations? Building footprints are surely simple enough that the advantages of vector representations for complex geometry don't really apply.
>
> > **Response sWPL-4** Rasterization, conceptually equivalent to aggregation of trivial vertices into buckets, does help in some cases (and is the traditional approach to handling vector shape data in ML thus far, with the exceptions noted in the literature review). Then the results will be critically dependent on augmentation, much more than here. Aggregation is also significantly dependent on the granularity of the raster imposed (and leads to artifacts, known in the geographical literature as the modifiable areal unit problem).

---

> > ### Comment · Reviewer_sWPL · 2023-11-23
> > **Thanks!**
> >
> > I thank the authors for their responses to my questions. However, I am unfortunately still not convinced the paper is above the acceptance bar.
> >
> > The author response still does not clarify why the method is designed in a way specific to simple polygons. I cannot see anything in the architecture specific to simple polygons and not to arbitrary graphs embedded in the plane -- bias towards polygons seems to come only from the training data. (By arbitrary graphs I did not mean graphs that have only topology and no embedding geometry. I meant graphs that are embedded in the plane, but are not necessarily polygons.)
> >
> > Re baselines, I would include raster baselines, and while such baselines would indeed require augmentation, synthetic augmentation (auto-rotate, reflect, translate, etc) is standard and easy to produce, and raster-based backbones are much better understood and likely more powerful. It is not at all clear to me that raster baselines would be worse than the proposed method in practice, even if they have occasional resolution shortcomings (the test shapes in the paper don't seem to have such high-frequency detail). There should be plenty of work on raster-based retrieval of glyphs or other silhouette images.
> >
> > I understand the point about simplifying collinear or non-informative vertices via Douglas-Puecker, but as in the other reviewer discussion, it is not clear whether DP can reliably converge to the same unique piecewise linear approximation of a curve starting from two very different oversegmented approximations, or whether the network is (largely) invariant to whatever it does converge to. Much more validation is needed to show that the method is robust to subdivision and discretization choices.

---

> > > ### Author Response · Authors · 2023-11-23
> > > **Response to Reviewer sWPL**
> > >
> > > Thank you for the clarification of your comments, this was not how we have interpreted your orignal questions.
> > > We *do* believe that our approach could also be used for the retrieval or what - in geospatial - is called LineStrings (general sequences of linear segements, but not closing a loop). There are fundamental differences that are not clear from pure vector encoding that distinguish closed linestrings, and polygons. These are *fundamentally* distinct in the geospatial field, and this is the main target for our method. The encoding is then *only* different by a constructor of the encoding (LineString, vs Polygon). They cannot be distinguished otherwise. We have therefore deliberately tried not to overstate the applicability of the method.
> > >
> > > Shape similarity is then not measurable by Hausdorff distance, but typically by Frechet distance, but again, only for non-closed linestrings. The qualitative assessment of a retrieval task could then be too ambiguous.
> > > We refrained from raster baselines, as they are critically dependent on resolution of the raster, and the contribution here is to enable what is sorely missing - the ability to undertake shape retrieval for e.g., building footprints directly on vector representations, a much more compact, and heavily used representation ( e.g., the vector tiels shown in Google Maps, and elsewhere).
> > >
> > > Finally, re Douglas Peuker - maybe we do not fully understand the comment *"it is not clear whether DP can reliably converge to the same unique piecewise linear approximation of a curve starting from two very different oversegmented approximations"*. DP does not converge, it is not a learning algorithm, but a purely deterministic, computational geometry approach for removal of uninformative vertices based on distance threshold, in an iterative manner. It will always result in the same outcome, for a given threshold.

---

> > > > ### Comment · Reviewer_sWPL · 2023-11-23
> > > > **Response to authors**
> > > >
> > > > Thanks for your comments. My point is simply that in assessing contributions to solving a target problem, it helps to look for the design decisions that directly address the specifics of the problem. In this case, the graph encoder is not specific to polygons, leaving open the possibility of better architectures that _do_ take advantage of the specifics of polygonal graphs. Otherwise, the authors get, for want of a better phrase, "less credit" for applying an unmodified generic approach.
> > > >
> > > > I disagree re the raster baseline. Building footprints seem particularly well-suited to rasterization, having simple, blocky structure and little high-frequency detail. Even a 512x512 raster would likely be enough in the vast majority of cases, differences finer than one pixel being at the level of natural noise. Many of the most successful retrieval algorithms on other vector representations (e.g. 3D meshes) operate via raster algorithms (e.g. see Su et al., "Multi-view Convolutional Neural Networks for 3D Shape Recognition", ICCV 2015). The storage representation need not constrain the retrieval representation -- I do not understand in which practical context a pure vector algorithm is "sorely missing".
> > > >
> > > > Re DP, sorry if that was unclear. By "convergence" I simply meant the result of applying DP to a point sequence. Of course it is deterministic (modulo degeneracies or symmetries). What I meant was that if the same canonical curve (say a circle) is approximated with two different piecewise linear sequences (which might naturally happen because of noise or human bias), and then you simplified each independently with DP, then the two simplified results would very likely not be identical, and your algorithm may or may not be robust to the differences.

---

> > > > > ### Author Response · Authors · 2023-11-23
> > > > > **Thank you!**
> > > > >
> > > > > Thank you - we appreciate your perspective, with respect to the framing of the contribution. Yet, we can only claim what we have tested and focused on, i.e. polygons.
> > > > >
> > > > > We reiterate that we have been fully focused on the retrieval task on vector geometries, not on rasterised images, as has been the tradition in CV for two decades. This is in that sense a departure from computer vision/remote sensing. Large vector datasets ( e.g., the recently released Microsoft Building footprints) are vector datasets, and so are many others ( not limited to building footprints, but this is a convenient polygonal focus area of relevance). There is currently no ability to undertake efficient ML on those shapes, *without rasterization* (we note that even the NUFT approach applied to vectors and used as baseline here *is* a rasterization approach). The resolution problem is simply fundamental to rasters ( as it is to vectors too, in a less overt way, where coordinate precision matters). Yet, here we really introduce the vector geometry retrieval problem, well translatable to any other vector polygonal shape ( and as you noted, possibly a non-polygonal graph).
> > > > >
> > > > > Regarding the DP point. Thank you for the clarification. Indeed, the approximation with linear segments of an arc or curve may result in distinct sequences. Hence this method here is for the retrieval of a "similar" shape, mimicking in that sense human perception. Distinct approximation may result after DP in different shapes, depending on how faithful and detailed the original approximation was done, and agreed upon. Importantly, however, we say robust, but not infallible. Under extreme simplification, all polygons may turn into a triangle... Here we focus on trivial vertices ( colinear ones, as well as those with very minor deviations from colinearity), that would not, perceptually, alter the shape.
> > > > >
> > > > > We trust that the reviewers will appreciate that we provide a thorough evaluation of our approach ( see new appendices, addressing all original comments of the reviewers). We are pleased with the discussion this triggered and would appreciate discussing it with the community at ICLR.

---

### Author Response · Authors · 2023-11-15
**Response to all reviewers**

We thank the reviewers for the highly encouraging reviews. We outline below how we understand the comments of the reviewers, and how we intend to address the comments in the brief rebuttal period. **We would appreciate if the reviewers were able to note whether we have understood their comments correctly.** We hope that our efforts will satisfy the reviewers’ questions and hope they may improve their ratings.

---

### Author Response · Authors · 2023-11-15
**References**

[Mai, Gengchen, et al.] Mai, Gengchen, et al. "Towards general-purpose representation learning of polygonal geometries." GeoInformatica 27.2 (2023): 289-340.

[Yan, Xiongfeng, et al.] Yan, Xiongfeng, et al. "Graph convolutional autoencoder model for the shape coding and cognition of buildings in maps." International Journal of Geographical Information Science 35.3 (2021): 490-512.

[Ai, Tinghua, et al.] Ai, Tinghua, et al. "A shape analysis and template matching of building features by the Fourier transform method." Computers, Environment and Urban Systems 41 (2013): 219-233.

[Saalfeld, Alan.] Saalfeld, Alan. "Topologically consistent line simplification with the Douglas-Peucker algorithm." Cartography and Geographic Information Science 26.1 (1999): 7-18.

[Lüscher, P., Weibel, R. and Burghardt, D.] Lüscher, Patrick, Robert Weibel, and Dirk Burghardt. "Integrating ontological modelling and Bayesian inference for pattern classification in topographic vector data." Computers, Environment and Urban Systems 33.5 (2009): 363-374.

[Arkin, Esther M., et al.] Arkin, Esther M., et al. An efficiently computable metric for comparing polygonal shapes. Cornell University Operations Research and Industrial Engineering, 1989.

---

### Author Response · Authors · 2023-11-23
**Final responses to Reviewers sWPL and SxCs**

Beyond the responses provided earlier, we now summarise the final edits to the paper. For brevity, we do not repeat the original questions or arguments raised in our original responses. We also appologise for the extremee brevity of some edits in the paper, due to paper length limitations.

**Response sWPL-1** We added the following wording to the Conclusions: "*The generalisation of CGAE to arbitrary graphs that only capture topology ( e.g., social networks) but not graph geometry or the application to ensembles of shapes defining semantics (terrace houses)(Luscher et al, 2009) are not addressed here.*"

**Response sWPL-2** Please, see Response **sEzw-1**

**Response sWPL-3** See response **SxCs-2** and **sEzw-4**.

**Response SxCs-1** Thanks for the suggestion. We update the manuscript by generating persistent diagrams for ten query geometries from the Melbourne dataset (as demonstrated in Appendix, Figure 22), showing the number of components (indicated by 0-th dimension topological features) and number of holes (indicated by 0-th dimension topological features). Note that the polygon geometries in Melbourne dataset (in total of 37139 samples) contains varying number of holes and are difficult to quantify as compared to other synthetic datasets (such as Glyph dataset and MNIST-cplx70k dataset in [Mai, Gengchen, et al.]).

**Response SxCs-2** Recall that our models are trained on the synthetic Glyph dataset and tested on real-world building footprints (OSM and Melbourne dataset). The number of vertex counts of the real-world building footprints vary compared to the synthetic dataset. We updated the manuscript by adding additional experiment results in Appendix, with Figures for the Melbourne Simplified datasets (Figures 13, 15), as well as the contrast to other benchmarks (NUFT, Figure 17, as well as Figures 19 and 21 for non-ML methods). These figures demonstrate the qualitative results of proposed CGAE over baselines on simplified polygon geometries (with trivial vertices removed) by the topology-preserving Douglas-Peucker algorithm.

**Response SxCs-4** We rephrase the sentence into "CGAE generalizes effectively the geometric information learned from simple polygons to complex shapes, demonstrating a desirable model property, i.e., decoupling of shape detail (i.e., polygon **trivial** vertex count.) from shape **retrieval** accuracy."

---

### Author Response · Authors · 2023-11-23
**Final responses to Reviewers R5mS and sEzw**

**Reviewer R5mS General Response** We included an algorithm statement with references to equations in Appendix C, including a listing of learnable parameters and their dimensions. We also include parameters of the training in Appendix B.

**Response R5mS-3** see Response **sWPL-1**

**Response R5mS-4** We thank Reviwer **R5mS** for pointing us to this paper. This is a good guidance for future work exploring implicit augmentations. Here, we apply explicit augmentations,  ratio $r$ = 20\%.

**Response R5mS-5** Losses were here all weighted equally. We make this now clear in conclusions, by adding: "multiple reconstruction losses and contrastive learning outperforms a baseline graph autoencoder with a single node reconstruction loss, as well as all baselines, **even without loss weight tunning**." to make it clear this is open to tweaks in applicaitons.


**Response R5mS-7** According to Open Geospatial Consortium. Opengis simple features specification for SQL revision 1.0, 2003, the linear ring is defined as “simple geometries [aka, without self intersections], where rings are defined by a series of points with linear interpolation between points, and the first and last point are identical.” We cite this standard in the papr, and edited the wording for clarity as "in the boundary linear ring of vertices"


**Response sEzw-1** We have provided comparisons with the two non-ML methods (Turning, and Procrustes). We note that in the study of baseline [Yan, Xiongfeng, et al.], similar non-ML FT-based methods [Ai, Tinghua, et al.] are already tested on the same OSM dataset as here, but show lesser performance.

**Response sEzw-2** We now include comparison with the NUFT-based baseline by [Mai, Gengchen, et al.]. We show the significantly superior performance of our methods in Tables 1-3, and qualitatively in the Appendix.

**Response sEzw-3** We note that the dataset preparation and testing of the present manuscript is fully documented in the reference provided as Anonymous. This manuscript is currently under double blind closed review. To mitigate the concerns of Reviewer **sEzw** regarding the issue of data preprocessing, we update the anonymous Figshare URL provided in Reproducibility Statement section, adding the implementation code of data preparation of three datasets (Glyph, OSM and Melbourne). We also note that there is no pre-alignement between the training dataset and the arbitrary rotations of buildings in the real world.

**Response sEzw-4** Please, refer to Response **SxCs-2**.

**Response sEzw-second round** We have rephrased the statement about our method being "free of vertex counts" in the Conclusion, to be consistent with the message in the paper overall, *"is robust to variable geometry vertex counts"*. We have added experiments demonstrating this robustness, significantly overperforming existent ML-based and non-ML baselines.

---

### Meta-Review · Area_Chair_Ntdn · 2023-11-30

**Metareview:**

The submission proposes a new approach, the Contrastive Graph Autoencoder (CGAE), to encode building footprint polygons using a message-passing graph encoder trained with node, edge and contrastive losses. CGAE outperforms traditional graph-based autoencoders (GAEs) through experiments with real-world building map datasets.

Strengths:
* reasonable tech contributions over Yan et al. e.g., retrieval of polygons with holes
* good results

Weaknesses:
* contributions are more applying a generic approach without insight to the specific problem.
* lack of important experiment results/limited comparison with baselines, the inclusion may change the story of the paper

**Justification For Why Not Higher Score:**

Out of four expert reviewers, three recommended rejection and one recommended acceptance. During the discussion phase, the positive reviewer thought that issues raised by other reviewers were convincing and also concurred with rejection. The AC found no reason to overturn the reviewers and decided rejection.

**Justification For Why Not Lower Score:**

NA

---

### Decision · Program_Chairs · 2024-01-16

Reject